# A Flexible Design Strategy for Three-Element Non-Uniform Linear Arrays

**DOI:** 10.3390/s23104872

**Published:** 2023-05-18

**Authors:** Andrea Quirini, Francesca Filippini, Carlo Bongioanni, Fabiola Colone, Pierfrancesco Lombardo

**Affiliations:** Department of Information Engineering, Electronics and Telecommunications DIET, Sapienza University of Rome, Via Eudossiana, 18, 00184 Rome, Italy

**Keywords:** non-uniform linear arrays, direction of arrival, passive location system, threshold region

## Abstract

This paper illustrates a flexible design strategy for a three-element non-uniform linear array (NULA) aimed at estimating the direction of arrival (DoA) of a source of interest. Thanks to the spatial diversity resulting from non-uniform sensor spacings, satisfactory DoA estimation accuracies can be achieved by employing a very limited number of receiving elements. This makes NULA configurations particularly attractive for low-cost passive location applications. To estimate the DoA of the source of interest, we resort to the maximum likelihood estimator, and the proposed design strategy is obtained by constraining the maximum pairwise error probability to control the errors occurring due to outliers. In fact, it is well known that the accuracy of the maximum likelihood estimator is often degraded by outliers, especially when the signal-to-noise power ratio does not belong to the so-called asymptotic region. The imposed constraint allows for the defining of an admissible region in which the array should be selected. This region can be further modified to incorporate practical design constraints concerning the antenna element size and the positioning accuracy. The best admissible array is then compared to the one obtained with a conventional NULA design approach, where only antenna spacings multiple of λ/2 are considered, showing improved performance, which is also confirmed by the experimental results.

## 1. Introduction

Several sensors aim at estimating the direction of arrival (DoA) of narrowband signals exploit linear antenna arrays with a small number of receiving elements. This choice allows for the minimizing of the number of receiving channels and the reducing of the overall system complexity, enabling the design of low-cost, lightweight, and compact sensors which are suitable for the mass-market.

Among the many applications that could benefit from a small number of receiving channels are low-cost radars for vehicular anti-collision applications—which estimate the direction of the target echo passive sonar systems—as well as passive radar (PR) systems [1], usually characterized by low-cost requirements compared to their active counterparts. Specifically, PRs based on different RF waveforms of opportunity have been widely studied. For example, considering the drone detection application, encouraging results have been obtained by parasitically exploiting a variety of RF waveforms, such as GSM [2], UMTS [3], LTE [4], DVB-T [5,6,7,8,9], and Wi-Fi signals [10,11].

Furthermore, low-cost sensors are particularly attractive in civil surveillance applications, e.g., PR sensors for the surveillance of private facilities [12], passive location sensors for the indoor monitoring of human activity [13], and sensors for the localization of survivors during the search and rescue operations after natural disasters. For all these applications, cost containment, limited complexity, and system lightness are crucial features. Therefore, the number of employed antenna elements and receiving channels must be kept as low as possible.

Three-element array configurations are currently widely considered for the practical, cost-effective implementation of limited beamforming and/or DoA estimation capability. As an example, the application for radio science experiments is presented in [14], whereas an example for 5G applications is reported in [15] for the C-band case using a uniform linear array (ULA) configuration with a pre-assigned element distance. Reference [16] presents the design and measurements of a 122 GHz on-chip patch antenna array in a 130 nm SiGe BiCMOS technology. The array is formed by three rectangular patches in the top metallization layer, with the ground plane in the lowest layer. Reference [17] presents a 300 GHz three-element on-chip patch antenna for applications in broadband Sub-THz communication and high-resolution radar sensing. The practical interest regarding the three-element array case also suggested the investigation of the possibility of removing the undesired e.m. coupling among its elements [18].

As pointed out in [19], when dealing with DoA estimation, two main reasons motivate the designer to use inter-element distances greater than *λ*/2, especially when using a small number of receiving channels:

Achieving a longer global array length, thus improving DoA estimation accuracy;Attaining compliance with technological constraints (e.g., the antenna element size might set a minimum inter-element distance).

Recently, significant efforts have been invested in the optimization of this simple, but useful, configuration. Unfortunately, it is well known that when estimating the DoA in an angular sector $\left[-\theta_{0},\theta_{0}\right]$, a ULA is affected by deterministic angular ambiguities if the uniform spacing between adjacent antenna elements exceeds $\lambda /2\left[1+\sin\theta_{0}\right]$. In particular, in reference [20], the binomial weight configuration is used, and both the element distance and the array factor are selected for a specific case.

In contrast, a non-uniform linear array (NULA) is only affected by statistical ambiguities, whose impact largely depends on the available signal-to-noise power ratio (SNR). The use of NULA configurations has been explored in [21,22].

Specifically, under very high SNR conditions, the NULA estimation accuracy reaches the Cramer–Rao bound (CRB), which essentially depends on the global array length. In this SNR region, known as asymptotic region, the estimation variance is inversely proportional to the SNR. However, below a specific SNR value, the presence of statistical ambiguities results in large estimation errors, also known as outliers, occurring with a non-negligible probability. Several authors addressed this issue in literature [23,24,25,26,27], exploring the use of different lower bounds, such as the Ziv-Zakai bound [27]. In this range of SNR values, known as the threshold region, the outlier probability monotonically grows as the SNR decreases, while the DoA estimation variance rapidly deviates from the CRB.

Therefore, a design strategy to select the optimal NULA configuration is particularly attractive, since the operational application of the previously introduced low-cost sensors typically faces limited SNR conditions. The design strategy proposed in [28] provides useful guidelines to select a proper array configuration. However, only antenna spacings in multiples of λ/2 are considered, which in turn leads to the following two weaknesses:

Depending on the SNR value and on the angular sector of interest, there is no guarantee that using inter-element distances with λ/2 quantization provides the optimal solution;Depending on the physical size of the employed antennas, operating with half-wavelength quantization may strongly limit the available design solutions, making the technological constraint more severe.

Moreover, it is unlikely that the quantized element spacing will allow for the investigation of the impact of the element positioning tolerance due to practical installation and/or manufacturing.

Array layout design approaches based on the symmetrical number system were proposed in [29,30], using an optimum symmetrical number system (OSNS) and a robust symmetrical number system (RSNS) of a pairwise relatively prime number, respectively. However, these design strategies do not provide a control of the performance in terms of statistical ambiguity resolution. More interestingly, an array spacing design procedure was presented in [31] for three- and four-elements arrays, including a constraint on a maximum admissible phase error in the design procedure, which allows for the exclusion of arrays that provide DoA ambiguity if the phase error exceeds a preassigned value. This approach can be considered as a first step towards our proposed design approach. In fact, in the following, we present an array spacing design procedure for the three-elements case that includes a direct control over the statistics of the ambiguity resolution. We fully characterize the probability of ambiguity resolution and provide an array spacing design procedure that guarantees a preassigned probability.

Specifically, based on the theoretical characterization of the DoA estimation error carried out by the author of [32], the proposed design strategy provides an array configuration which is not subject to the quantization constraint on the inter-element distance. The findings in [32] allow us to devise a procedure that constrains the searched NULA array configurations to provide an outlier probability below a maximum tolerable value. Moreover, the prediction of the maximum likelihood estimator (MLE) performance provided therein, in both the threshold and asymptotic regions, enables the identification of the solution with minimum mean square error (MSE), even for limited values of the available SNR. This also allows us to include additional constraints related to practical array manufacturing. It is interesting to note that design approaches for arrays with more than three elements were proposed in [33,34] as evolutions or combinations of the three-elements array procedures. Therefore, the contribution of this paper is also expected to provide useful elements for the design of longer arrays.

The remainder of the paper is organized as follows. In Section 2, the signal model, DoA MLE, and theoretical characterization of the MSE are introduced. In Section 3, the admissible pairwise probability (APP) region is defined as the parameters region containing the arrays that satisfy the constraint on the maximum outlier probability for a chosen SNR. Based on the APP region, the design strategy is derived as a constrained optimization problem and is presented in Section 4. In Section 5, a numerical analysis is carried out to assess the performance of various NULA configurations obtained with the proposed strategy. A comparison with the design strategy proposed in [27] is also carried out to show the limitations of the quantization constraint. Section 6 shows the benefits of the proposed strategy over the conventional approach when adding technological constraints, which further limits the available array configurations. Finally, in Section 7, the advantages shown for the simulated data are tested against an experimental dataset collected in the Wi-Fi band. Concluding remarks are reported in Section 8.

## 2. Array Signal Model and Performance Prediction

### 2.1. Array Geometry and Signal Model

Consider an N-element linear array receiving a narrowband signal from a single source with DoA $\theta_{s}$, measured relative to the array boresight. The displacements $d_{n},\;n=0,\ldots ,\;\;N-1$ of the element positions compared to the first element are collected into vector $z=\left[0\;d_{1}\;\ldots d_{N-1}\right]$. The described array geometry is sketched in Figure 1.

Let s(u) be the steering vector, defined as
(1)$$s\left(u\right)=\left[\begin{array}{cccc} 1 & e^{\left(j\frac{2\pi }{\lambda }d_{1}u\right)} & \cdots & e^{\left(j\frac{2\pi }{\lambda }d_{N-1}u\right)} \end{array}\right]$$
where λ is the wavelength, and u=sinθ is the steering direction. After I/Q demodulation to baseband, the N received signals are sampled and digitized. The snapshot of the array, collected at a specific time $t_{0}$, is represented by the N×1 complex vector x. Since the receiving elements are affected by thermal noise, the complex array output after down-conversion, filtering, and sampling can be arranged into the N-dimensional column vector
(2)$$x=As\left(u_{s}\right)+n.$$

Assuming a deterministic signal model as in [32,35]:

A is the generic sample of the complex baseband source signal and is modeled as a deterministic, but unknown, parameter.$S(u_{s})$ is the target steering vector, obtained by Equation (1) along the direction of the source, namely $u_{s}=\sin(\theta_{s})$.n is an N-dimensional column vector collecting the noise samples at the receiver. 

Based on the pattern of the employed antenna elements and the possible presence of additional electromagnetic shielding, we assume that the antenna array can only receive signals with DoA in the sector $U_{max}=\left[-u_{max},u_{max}\right]$, being $u_{max}=\sin\left(\theta_{max}\right)$. This angular sector is represented by the green region in Figure 1. Noise samples at the N antenna elements are assumed to be distributed according to an independent, zero-mean complex Gaussian random variable, with the same mean square value $\sigma_{n}^{2}$, which is also independent of the source signal.

### 2.2. Maximum Likelihood DoA Estimation

To achieve the best estimation of the source DoA, we resort to the MLE. Based on the model in Equation (2), this is obtained by looking for the value of u that maximizes the likelihood function
(3)$$V\left(u\right)={\left|s^{H}\left(u\right)x\right|}^{2}$$

Looking for the maximum in the region $u\in U_{max}$, the ML estimate is given by:(4)$${\hat{u}}_{s}=\underset{u\in U_{\max}}{\mathrm{argmax}}\left\{V\left(u\right)\right\}$$

The estimate ${\hat{\theta }}_{s}$ of the DoA $\theta_{s}$ is then obtained by inverting the relationship $u_{s}=\sin\left(\theta_{s}\right)$.

Depending on the application at hand, the sources of interest may be only those with a DoA that lies inside the angular sector defined by $U_{int}=\left[-u_{int},u_{int}\right]$, being $u_{int}=\sin\left(\theta_{int}\right)$, possibly narrower than $U_{max}$ (namely $\theta_{int}\leq \theta_{max}$), as represented by the red region in Figure 1. Therefore, estimates that fall outside $U_{int}$ can be discarded as not being of interest. This allows for the removal of potential outliers of the ML estimation that correspond to largely displaced DoA. It is further noticed here that there might be a specific interest to only optimize the estimation performance inside an even narrower angular sector $U_{opt}$ represented by the yellow region in Figure 1, which will be discussed in the following sections of this paper.

### 2.3. Performance Evaluation and MSE Approximation

The design strategy presented in the following sections of this paper aims at selecting array configurations with the best possible performance. A basic building block for this procedure is provided by a robust performance prediction of the DoA estimation accuracy, evaluated in terms of MSE. A vast amount of technical literature has addressed the characterization of this MSE, which depends both on the array configuration and on the signal parameters.

Figure 2 qualitatively shows the behavior of the estimation error: the continuous blue line represents the MSE, while the dashed blue line represents the CRB.

As shown in Figure 2, three different operative regions can be identified, which can be explained as follows. In the absence of noise, we have an ideal value of the likelihood function
(5)$$L\left(u\right)={\left|s^{H}\left(u\right)s\left(u_{s}\right)\right|}^{2},$$

This provides the array beampattern for a source coming from $u_{s}$, which always exhibits a maximum in the true target DoA, and has a mainlobe width that solely depends on the effective array length. For N>2, L(u) is characterized by sidelobes, whose positions $u_{m},\;m=1,2,\ldots$, levels, and widths depend on the array geometry (i.e., on the inter-element distances in z).

When the SNR is very high, there is a negligible probability that the noise values added to the signal source snapshot provide a peak of V(u) in the direction $u_{m}$ of a sidelobe of L(u), namely outside its main lobe, rather than in the direction of the true DoA, $u_{s}$. In other words, in such conditions, thermal noise causes only slight variations in the peak location inside the main lobe, while the outlier probability is negligible. Therefore, the MSE of the ML DoA estimate accurately attains the CRB, derived as in [36,37,38,39]
(6)$$CRB=\frac{1}{8\pi^{2}SNR\sum_{n=0}^{N-1}{\left(\frac{d_{n}}{\lambda }-\frac{1}{N}\sum_{p=0}^{N-1}\frac{d_{p}}{\lambda }\right)}^{2}},$$

The CRB in Equation (6) only depends on the width of the main lobe, namely on the global array length, and is inversely proportional to the SNR. This region is referred to as the asymptotic region.

As the SNR decreases, the MSE rapidly deviates from its asymptotic behavior. Moreover, the probability that thermal noise results in a maximum of V(u) in correspondence of a sidelobe of L(u) becomes non-negligible. This leads to a DoA estimate well outside the main lobe of the likelihood function (namely an outlier), which results in a sensibly larger MSE. Therefore, the expression of the CRB can no longer be used as an approximation of the MSE. This threshold region is of major interest for practical applications, since in most cases, low-cost sensors need to operate with limited SNR values.

Finally, for very low SNR values, the probability that the effect of thermal noise provides a peak of V(u) in the sidelobes of L(u) becomes so high that the estimation process is undermined by the presence of several large errors. Under such conditions, the DoA estimates are uniformly distributed across the entire parameter space, and the MLE does not provide any reliable information on the true source DoA. This region is referred to as the no-information region.

Based on the discussion above, the authors of [32,35], derived an approximation of the MSE, denoted as ζ, achieved with the MLE in both the threshold and asymptotic regions: (7)$$\zeta \left(SNR,d_{1},\ldots ,\;d_{N-1},u_{s}\right)≜E\left[{\left({\hat{u}}_{s}-u_{s}\right)}^{2}\right]\approx \left[1-\overset{M}{\underset{m=1}{\sum }}P_{m}\right]\cdot CRB+\overset{M}{\underset{m=1}{\sum }}{\left(u_{m}-u_{s}\right)}^{2}P_{m},$$
where

M represents the number of sidelobe peaks in the angular sector $U_{int}$, where the estimates are considered of interest and are retained;$P_{m}=P_{m}\left(SNR,d_{1},\;\ldots ,\;d_{N-1},u_{s}\right)$ represents the mth pairwise error probability, i.e., the probability that the mth sidelobe peak in position $u_{m}$ is chosen by the MLE search instead of the main lobe peak.

Notice that the union bound approximation derived in [40] and used in [32] ignores the events of two or more sidelobe peaks simultaneously being higher than the main lobe peak. Based on this, the probability of outlier $P_{out}$ can be approximated as
(8)$$P_{out}≅\overset{M}{\underset{m=1}{\sum }}P_{m}=\overset{M}{\underset{m=1}{\sum }}Pr\left\{V\left(u_{m}\right)>V\left(u_{s}\right)\right\}.$$

Under a deterministic model assumption for the signal amplitude, a closed form expression for the pairwise error probability was also derived in [32] based on the discussion in [41], obtaining:(9)$$P_{m}=Q\left(a,b\right)-\frac{1}{2}e^{-\frac{SNR}{2}}I_{0}\left(\frac{SNR}{2K}\sqrt{L(u_{m})}\right),$$
where Q(a,b) is the Marcum *Q*-function, $I_{0}\left(\cdot \right)$ is the modified Bessel function of the first kind and order 0, and its arguments depend on the array beampattern evaluated at the m-th sidelobe peak, located in $u_{m}=\sin\left(\theta_{m}\right)$:(10)$$a≜\sqrt{\frac{SNR}{2N}\left(N-\sqrt{N^{2}-L(u_{m})}\right)},\;\;b≜\sqrt{\frac{SNR}{2N}\left(N+\sqrt{N^{2}-L(u_{m})}\right)},$$

In the remainder of the paper, we build upon this theoretical performance characterization to derive a flexible design strategy for a three-element NULA.

## 3. Admissible Pairwise Probability Region

From the approximation in Equation (7), we notice that the values of $P_{m}$ act as scaling factors for the large error contributions ${\left(u_{m}-u_{s}\right)}^{2}$ that add to the CRB to provide the global MSE. Therefore, the array configuration achieving the minimum MSE certainly lies in a region where the pairwise probabilities $P_{m}$ have negligible values. Consequently, an important step toward the array optimization consists of the identification of the region of the array parameters, namely $d_{n},\;\;n=1,\;\ldots ,\;\;N-1$, where all $P_{m}$ values are below an assigned threshold (i.e., the maximum acceptable value). This region will be referred to in the following as the APP region.

To identify the APP region, we notice that the mth pairwise error probability in Equation (9) is a monotonic decreasing function of both the SNR and the array beampattern $L(u_{m})$ evaluated at the mth sidelobe peak location $u_{m}$. Therefore, the array configuration achieving the minimum MSE depends on the specific SNR conditions. To carry out the optimization, we first select a fixed value for the SNR, i.e., $SNR_{0}$. With this choice, the constraint on $P_{m}\leq \;P_{max}$ is equivalent to setting an upper bound on the value of L(u), namely $L\left(u\right)\leq L_{max}$, where $L_{max}=L_{max}(SNR_{0},P_{max}$).

By using the steering vector in Equation (1), the likelihood function L(u) in Equation (5) can be expressed in terms of the array parameters $d_{n},\;n=1,\;\ldots ,\;N-1$:(11)$$L\left(u\right)={\left|\overset{N-1}{\underset{k=0}{\sum }}e^{j2\pi \Delta_{n}}\right|}^{2}\leq L_{max},$$
where
(12)$$\Delta_{n}≜\frac{d_{n}}{\lambda }\left(u-u_{s}\right),\;\;n=0,\;\ldots ,\;N-1.$$

While this constraint is valid for any value of N, we are especially interested to the case of N=3, which is appealing when addressing the design of low-cost, lightweight sensors, as described in the Section 1. For N=3 array elements, the constraint in Equation (11) can be simplified as:(13)$$L\left(\Delta_{1},\Delta_{2}\right)={\left|1+e^{j2\pi \Delta_{1}}+e^{j2\pi \Delta_{2}}\right|}^{2}\leq L_{max},$$
which is a function of two parameters and can be easily represented in a plane. Figure 3 shows a contour plot representing the function $L\left(\Delta_{1},\Delta_{2}\right)$ on a $\Delta_{1},\;\Delta_{2}$ axis system.

Notice that $L\left(\Delta_{1},\Delta_{2}\right)$ is a periodic function of both $\Delta_{1}$ and $\Delta_{2}$ with a unity period, i.e., $L\left(\Delta_{1}-k,\Delta_{2}-h\right)=L\left(\Delta_{1},\Delta_{2}\right)$ for any h,k∈ℤ. For future reference, we define its single period $L_{0}$ as
(14)$$L_{0}\left(\Delta_{1},\Delta_{2}\right)=\{\begin{array}{c} L\left(\Delta_{1},\Delta_{2}\right),\;\;-\frac{1}{2}<\Delta_{1},\Delta_{2}\leq \frac{1}{2}\;\\ 0,\;\;\;\;\;\;\;\;\;\;\;\;\;\;\;\;elsewhere \end{array}.$$

This is represented in the red box in Figure 3, where the numeric labels refer to the contour lines drawn for different possible values of $L_{max}$ (with $L_{max}\leq N^{2}=9)$. Using the definition of $L_{0}\left(\Delta_{1},\Delta_{2}\right)$, we can rewrite $L\left(\Delta_{1},\Delta_{2}\right)$ as
(15)$$L\left(\Delta_{1},\Delta_{2}\right)=\overset{\infty \;}{\underset{h,k=-\infty }{\sum }}L_{0}\left(\Delta_{1}-k,\Delta_{2}-h\right),$$
so that the replica for *k*, *h* = 0 represents the main lobe of the ideal likelihood function, while the replicas for k,h≠0 represent its grating lobes.

Our goal now is to exploit the constraint on the likelihood function $L\left(\Delta_{1},\Delta_{2}\right)\leq L_{max}$ to identify the subset of array configurations satisfying the constraint on the maximum pairwise error probability, under the assumed condition of $SNR=SNR_{0}$. 

To achieve this, we first represent the beampattern L(u) of a specific array configuration $z=\left[0,\;d_{1},d_{2}\right]$ in the plane $\left(\Delta_{1},\Delta_{2}\right)$. From Equation (12), we notice that $\Delta_{1}$ and $\Delta_{2}$ depend both on the array design parameters $d_{1}$ and $d_{2}$ and on the difference $\left(u-u_{s}\right)$. However, the ratio between $\Delta_{2}$ and $\Delta_{1}$ is independent of $\left(u-u_{s}\right)$, and it is equal to the ratio between $d_{2}$ and $d_{1}$, namely $\alpha =\frac{d_{2}}{d_{1}}=\frac{\Delta_{2}}{\Delta_{1}}$. Therefore, as the steering direction u changes, the locus of points corresponding to the array beampattern L(u) of a specific 3-element array $z=\left[0,\;d_{1},\alpha d_{1}\right]$ can be represented in the plane $\left(\Delta_{1},\Delta_{2}\right)$ as a linear segment $\bar{NP}$ with slope α. For a specific array configuration, the endpoints N and P of the segment $\bar{NP}$ only depend on the maximum and minimum values assumed by the difference $\left(u-u_{s}\right)$. By recalling that the array can only receive signals with DoA $u_{s}$ in the sector $U_{max}=\left[-u_{max},u_{max}\right]$, and that the we are only interested in estimating the DoA of the sources located in the sector $U_{int}=\left[-u_{int},u_{int}\right]$, we have $u_{s}\in U_{max}$ and $u\in U_{int}$. Hence, the difference $u-u_{s}$ is bounded as
(16)$$-\mu_{max}<u-u_{s}<\mu_{max},$$
where $\mu_{max}≜u_{max}+u_{int}$. For a given $\mu_{max}$, the coordinates of the endpoints N and P for the array z in the plane $\left(\Delta_{1},\Delta_{2}\right)$ can be derived by inverting Equation (12)
(17)$$\{\begin{array}{c} N=\left(-\frac{d_{1}}{\lambda }\mu_{max},-\frac{\alpha d_{1}}{\lambda }\mu_{max}\right)\\ P=\left(+\frac{d_{1}}{\lambda }\mu_{max},+\frac{\alpha d_{1}}{\lambda }\mu_{max}\right)\; \end{array}.$$

Given the biunivocal relationship between z and $\bar{NP}$, determining whether an array configuration is admissible or not is now straightforward. Specifically, an array z can be considered admissible only if the corresponding segment in the $\left(\Delta_{1},\Delta_{2}\right)$ plane does not intersect the contour $L_{0}\left(\Delta_{1}-k,\Delta_{2}-h\right)=L_{max}$ for any k,h≠0. This condition guarantees that the sidelobes of the array beampattern L(u) are lower than $L_{max}$, which in turn allows for satisfying the constraint on the maximum pairwise error probability. The following example allows for the visualization of the relationship between L(u) and $\bar{NP}$, and further clarifies the notion of admissibility. 

Let us assume an $SNR_{0}=15\;\mathrm{dB}$ and a maximum acceptable pairwise probability $P_{max}=10^{-5}$. By inverting Equation (9), we obtain $L_{max}\approx 5.8612$. The corresponding contour plot of the $L\left(\Delta_{1},\;\Delta_{2}\right)$ function is represented in Figure 4a. As is apparent, the region external to the contour lines, namely the green area, represents the set of $\left(\Delta_{1},\Delta_{2}\right)$ values that satisfy the constraint $L\left(\Delta_{1},\Delta_{2}\right)\leq L_{max}$.

Now, consider the following array configurations, characterized by α=2.1: (18)$$.\;\{\begin{array}{c} \begin{array}{c} \begin{array}{c} z_{1}=\rho \;\left[0\;0.150\;0.315\right]\lambda \\ z_{2}=\rho \;\left[0\;0.300\;0.630\right]\lambda \end{array}\\ z_{3}=\rho \;\left[0\;0.450\;0.945\right]\lambda \end{array}\\ z_{4}=\rho \;\left[0\;0.600\;1.260\right]\lambda \end{array}$$

As explained previously, these array configurations can be represented in the plane $\left(\Delta_{1},\Delta_{2}\right)$ as segments lying on the line $\Delta_{2}=2.1\Delta_{1}$, and the extrema of these segments depend on the value of μ. Assuming $\mu_{max}=2$ (i.e., $u_{max}=u_{int}=1$), we evaluate the endpoints N and P through Equation (17), and we represent the four considered configurations in Figure 4a as colored segments $\bar{NP}$ to verify their admissibility.

As visible in Figure 4b, for ρ=1, configurations $z_{1}$ and $z_{2}$ are admissible, since the segments representing their beampatterns do not intersect any replica of the likelihood function (except for the replica k=h=0, which is the likelihood function main lobe). Conversely, configurations $z_{3}$ and $z_{4}$ are not admissible, since the segments representing their beampatterns do intersect the contour $L_{0}\left(\Delta_{1}-k,\Delta_{2}-h\right)=L_{max}$ for (k,h)=(1,2) and for (k,h)=(−1,−2). The four array beampatterns are also represented in Figure 5 as a function of $\left(u-u_{s}\right)$. Comparing these beampatterns, we notice that the array configurations denoted as $z_{1}$ and $z_{2}$ achieve the lowest sidelobes, while the others have high sidelobes that prevent fulfilling the requirement on $L_{max}$.

Notice how the beampatterns of the array configurations characterized by the same α value are merely stretched versions of the same likelihood function. Specifically, the slope $\alpha =d_{2}/d_{1}$ defines the shape of the likelihood function, while the value μ determines the abscissa $\frac{d_{1}}{\lambda }\mu_{max}$ of the $\Delta_{1}$ axis in which the likelihood function is cut to obtain the array beampattern.

As mentioned in Section 1, a straightforward approach to measure the robustness of a given array configuration to estimation ambiguities was proposed in [31], based on the maximum admissible phase error. Therein, the authors introduced the synthetic parameter $min\left(S_{pq}\right)$, defined as the minimum distance between the folded replicas of the line $\phi_{2}=\alpha \phi_{1}$, with the wrapped phase within the angular sector Φ=[0°,360°]. The minimum distance $min(S_{pq}$) of an array provides its maximum admissible phase error and is therefore a measure of the array’s immunity to ambiguities. By assigning a minimum acceptable distance, the parameter $min\left(S_{pq}\right)$ can be used to determine the admissibility of an array.

The approach in [31] is inherently different from the one proposed in this work, since it capitalizes on the phase ambiguity of the involved baselines rather than on the statistical formulation of the MLE. However, the two approaches generally provide consistent results. To carry out a comparison, we used the algorithm described in [31] to evaluate $min\left(S_{pq}\right)$ for the array configurations in Figure 4a, with size scaled by ρ=0.7, 0.8, 0.9, 1.0, 1.1, 1.2, 1.3, and reported their values, together with the corresponding maximum pairwise error probability $\underset{m}{\max}P_{m},\;\mathrm{in}\;\mathrm{Table}\;1\;$.

As visible in Table 1, requiring that $P_{m}\leq P_{max},\forall m$ is equivalent to requiring that $min\left(S_{pq}\right)>S_{min}$. However, while $\underset{m}{\max}P_{m}$ ranges from $10^{-20}$ to 0.2, taking several intermediate values for different array configurations, $min\left(S_{pq}\right)$ only takes two values, namely 170.25 for more robust configurations, and 15.48 for less robust examples. This suggests that constraining $min\left(S_{pq}\right)$ provides an approximate admissibility criterion, while the proposed approach enables a more accurate assessment of array robustness to outliers. As a consequence, the two approaches provide consistent results for all array configurations, showing a well-defined behavior with respect to the presence of outliers. In contrast, they might yield conflicting indications for edge case configurations. 

For example, from Table 1, we notice that the constraint $P_{m}\leq P_{max}=10^{-5},\forall m$ is almost equivalent to the constraint $min\left(S_{pq}\right)>S_{min}=15.48$. With these parameters, the two approaches agree on the admissibility of both configurations $z_{1}$ and $z_{2}$ for any ρ value. However, when ρ=0.7, the criterion $min\left(S_{pq}\right)>15.48$ suggests that all the included array configurations should be considered admissible. Conversely, we notice that configuration ***z***_4_ does not satisfy the constraint $P_{m}\leq 10^{-5},\forall m$, being characterized by $\underset{m}{\max}P_{m}=2\cdot 10^{-5}$, and thus it would be discarded using the proposed approach. Similar considerations apply to configuration $z_{3}$ when ρ=1.

Clearly, different values of SNR and $P_{max}$ can lead to different edge cases on these arrays. Furthermore, additional edge cases can be found, considering array configurations with different slopes α. However, the example above shows that the ML-derived proposed approach enables a more precise assessment of the array robustness to outliers, allowing us to better identify the boundaries of the APP region where the best constrained configurations in terms of MSE typically lie. This is shown in the following.

The green region in Figure 4b identifies the subset of abscissa values $\frac{d_{1}}{\lambda }\mu_{max}$, for which a cut of the likelihood function provides an admissible array beampattern, namely L(u), such that $L\left(u\right)\leq L_{max}$. The extrema of the green patch are denoted as $\pm \Delta_{1}^{max}$. This abscissa value defines the longest admissible array with α=2.1, so that every array configuration $z=\left[0,\;d_{1},\alpha d_{1}\right]$ such that $\frac{d_{1}}{\lambda }\mu_{max}\leq \Delta_{1}^{max}$ is admissible.

Notice that the value of $\Delta_{1}^{max}$ depends on the chosen slope α. Hence, if we determine the value $\Delta_{1}^{max}\left(\alpha \right)$ of maximum abscissa for every possible slope α, we can identify the region in the plane $\left(\Delta_{1},\Delta_{2}\right)$ which contains every array configuration that satisfies $L\left(u\right)\leq L_{max}$, namely the APP region. 

In principle, the value $\Delta_{1}^{max}\left(\alpha \right)$ can be derived as the intersection between the contours $L_{0}\left(\Delta_{1}-k,\Delta_{2}-h\right)=L_{max}$ and a line through the origin with slope α. Thus, the APP region can be derived in the plane $\left(\Delta_{1},\Delta_{2}\right)$ before setting a specific value for $\mu_{max}$.

Once the APP region has been identified, the value of $\mu_{max}$ is required to relate the coordinates $\left(\Delta_{1},\Delta_{2}\right)$ to the inter-element distances $\left(d_{1},d_{2}\right)$. In other words, as visible from Equation (17), the definition of the endpoints of the segment $\bar{NP}$ associated to an array z requires both the knowledge of $\left(d_{1},d_{2}\right)$ and of $\mu_{max}$. Therefore, the admissibility of a specific array z depends not only on the constraint on the pairwise error probability, but also on how the sectors $U_{max}$ and $U_{a}$ are chosen.

For every slope α, the analytical value of the optimal $\Delta_{1}^{max}\left(\alpha \right)$ can be identified as follows. First, we notice that $L_{0}\left(\Delta_{1},\Delta_{2}\right)=3+2\cos\left(2\pi \Delta_{1}\right)+2\cos\left(2\pi \Delta_{2}\right)+2\cos\left[\left(2\pi (\Delta_{2}-\Delta_{1}\right)\right]$ is symmetrical with respect to the two bisectors of the first and the second quadrants of the plane $\left(\Delta_{1},\Delta_{2}\right)$.

By defining the new set of coordinates γ and δ, such that:(19)$$\{\begin{array}{c} \Delta_{1}=\frac{\delta -\gamma }{2\pi }\\ \Delta_{2}=\frac{\delta +\gamma }{2\pi } \end{array}.$$
and applying the transformation, we obtain
(20)$$L_{0}\left(\gamma ,\delta \right)=1+4\cos(\delta )\cos(\gamma )+4\cos^{2}(\gamma )=L_{max},$$
which can be easily solved for δ, obtaining:(21)$$\delta =\pm acos\left\{\frac{L_{max}-1}{4cos(\gamma )}-\cos\left(\gamma \right)\right\}.$$

Using Equation (21) in Equation (19) allows us to express the points on the contour of $L_{0}\left(\Delta_{1},\Delta_{2}\right)$ defined by $L_{max}$, in terms of γ, as:(22)$$\{\begin{array}{c} \Delta_{1}=\Delta_{1}\left(\gamma \right)=\pm \frac{1}{2\pi }\mathrm{acos}\left\{\frac{L_{max}-1}{4\cos(\gamma )}-\cos(\gamma )\right\}-\frac{\gamma }{2\pi }\\ \begin{array}{c} \Delta_{2}=\Delta_{2}\left(\gamma \right)=\pm \frac{1}{2\pi }\mathrm{acos}\left\{\frac{L_{max}-1}{4\cos(\gamma )}-\cos(\gamma )\right\}+\frac{\gamma }{2\pi } \end{array} \end{array}.$$

Since the segment representing an array lies on a line through the origin, the slope of the line crossing the contour of the replica (k,h) is given by the ratio $\alpha_{k,h}\left(\gamma \right)=\frac{\Delta_{2}\left(\gamma \right)+h}{\Delta_{1}\left(\gamma \right)+k}$
(23)$$\{\begin{array}{c} \alpha_{k,h}^{+}\left(\gamma \right)=\frac{\frac{1}{2\pi }\mathrm{acos}\left\{\frac{L_{max}-1}{4\cos(\gamma )}-\cos(\gamma )\right\}+\frac{\gamma }{2\pi }+h}{\frac{1}{2\pi }\mathrm{acos}\left\{\frac{L_{max}-1}{4\cos(\gamma )}-\cos(\gamma )\right\}-\frac{\gamma }{2\pi }+k}\;\;\;\;\\ \begin{array}{c} \alpha_{k,h}^{-}\left(\gamma \right)=\frac{\frac{1}{2\pi }\mathrm{acos}\left\{\frac{L_{max}-1}{4\cos(\gamma )}-\cos(\gamma )\right\}-\frac{\gamma }{2\pi }-h}{\frac{1}{2\pi }\mathrm{acos}\left\{\frac{L_{max}-1}{4\cos(\gamma )}-\cos(\gamma )\right\}+\frac{\gamma }{2\pi }-k} \end{array} \end{array},$$
where $\alpha_{k,h}^{+}\left(\gamma \right)$ refers to the slope obtained from the positive solutions, and $\alpha_{k,h}^{-}\left(\gamma \right)$ refers to the slope obtained from the negative ones. As γ varies in the range (−2π,2π), the results in Equation (23) provide the slopes of the lines crossing the replica (k,h) of the contour $L_{0}=L_{max}$ and allow us to identify the maximum and minimum slope of its crossing line
(24)$$\{\begin{array}{c} \alpha_{k,h}^{min}\left(\gamma \right)=\min\left\{\alpha_{k,h}^{+}\left(\gamma \right),\alpha_{k,h}^{-}\left(\gamma \right)\right\}\\ \begin{array}{c} \alpha_{k,h}^{max}\left(\gamma \right)=\max\left\{\alpha_{k,h}^{+}\left(\gamma \right),\alpha_{k,h}^{-}\left(\gamma \right)\right\} \end{array} \end{array}.$$

By exploiting the previous analytical re41sults, together with the observation that the lower order replicas block the limiting effect of the higher order replicas, we propose the following procedure to identify the APP region in the $\left(\Delta_{1},\Delta_{2}\right)$ plane. First, we identify the $\Delta_{1}^{max}\left(\alpha \right)$ relative to the slopes crossing the lowest order replica, then we progressively evaluate the $\Delta_{1}^{max}\left(\alpha \right)$ relative to the slopes crossing higher order replicas, until all slopes have been assigned a value $\Delta_{1}^{max}\left(\alpha \right)$, since by construction $d_{2}>d_{1}$, in the following, we only consider α>1. The following procedure is obtained:

Inside the set of all possible slopes (1,+∞), determine $\Delta_{1}^{max}\left(\alpha \right)$ for the subset of slopes α blocked by the first-order replica of the $L\left(\Delta_{1},\;\Delta_{2}\right)$ function, i.e., (k,h)=(0,1). Notice that all the replicas (k,h)=(0,h) are blocked by replica (0,1) so they will be ignored in the following.Determine the set of the non-blocked slopes after step I.Determine $\Delta_{1}^{max}\left(\alpha \right)$ for the subset of slopes α blocked by next higher-order replica of the $L\left(\Delta_{1},\;\Delta_{2}\right)$ function, i.e., (k,h)=(1,1). Notice that the replicas (k,h)=(h,h) are blocked by replica (1,1) so they will be ignored in the following.Determine the set of the non-blocked slopes after step III.Determine the subset of slopes α blocked by next higher-order replica of the $L\left(\Delta_{1},\;\Delta_{2}\right)$ function.Determine the set of the non-blocked slopes after step V.Repeat steps V and VI until the set of non-blocked slopes is empty.

Conceptually, the presented procedure could be extended to *N* = 4 antenna elements, by operating in a 3D space, where the contours are replaced by surfaces, and the APP region resembles a Swiss cheese block. While the extension is straightforward, the analytical derivation would make the description uselessly cumbersome. Therefore, we avoid an explicit illustration of this case and leave it to the purview of interested reader.

As an example of the application of the N=3 case, the proposed procedure is applied for the case with SNR=20dB and $P_{max}=10^{-4}$, resulting in $L_{max}=8.1739$. Since only slopes α>1 are of interest, the APP region occupies only the portion above the bisector of the first and third quadrant of the plane $\left(\Delta_{1},\Delta_{2}\right),\;$ as shown in Figure 6, after remapping in the plane $\left(\Delta_{1},\Delta_{2}-\Delta_{1}\right)$.

Notice that the above procedure allowed us to determine the APP region in the plane $\left(\Delta_{1},\Delta_{2}\right)$. By defining the extrema $u_{max}$ and $u_{int}$ of the angular sectors $U_{max}$ and $U_{int}$, we can map the admissible region in the plane of the array parameters $\left(d_{1},d_{2}\right)$. The identified APP region will be the basis for the array design approach presented in the following section.

## 4. Three-Element NULA Design Strategy

In this section, we present the design strategy for the three-elements array aimed at low-cost low-weight sensors for DoA estimation. Based on the MSE approximation in Equation (7), it is clear that minimizing the CRB does not guarantee the best performance outside the asymptotic region, since for limited SNR, there might be a significant error contribution caused by the outliers that tend to concentrate around the DoAs, $u_{m}$, where the beampattern exhibits its peaks. Moreover, Equation (7) shows that the values of $P_{m}$ act as scaling factors for the large error contributions ${\left(u_{m}-u_{s}\right)}^{2}$ that add to the CRB.

Therefore, for a given $SNR_{0}$, the only configurations that are of potential interest (i.e., provide small MSE) all have $P_{m}$ values below the maximum acceptable value, *P_max_*, and belong to the APP region. The procedure presented in Section 3 is therefore a useful tool to restrict the search for valuable array configurations and is used as an important part of the proposed design approach.

Inside the APP region, the optimum array configuration $z=\left[0,\;d_{1},\;\ldots ,\;d_{N-1}\right]$ is obtained, adopting a minimax criterion as the one that minimizes the maximum MSE over any desired range of source angles, $U_{opt}=\left[-u_{opt},u_{opt}\right],\;u_{opt}=\sin\left(\theta_{opt}\right)$:(25)$$P\{\begin{array}{c} \underset{d_{1},\ldots ,d_{N-1}}{\mathrm{argmin}}\;\;\left\{\underset{u_{s}\in U_{opt}}{\max}\;\;\zeta \left(SNR_{0},d_{1},\;\ldots ,\;d_{N-1},u_{s}\right)\right\}\;\\ \begin{array}{c} s.t.\;\mathrm{array}\;z=\left[0,\;d_{1},\;\ldots ,\;d_{N-1}\right]\in APP\;\mathrm{region}\;\mathrm{for}\;\mathrm{assigned}\;SNR_{0},\;\theta_{max},\;\theta_{int},P_{max} \end{array} \end{array}.$$

$U_{0}$ can be either coincident with the whole sector of sources of interest $U_{a}$ (i.e., $\theta_{opt}=\theta_{int}$), or it can be a smaller sector, where a higher accuracy is desired (i.e., $\theta_{opt}<\theta_{int}$). This formulation is equivalent to:(26)$$P\{\begin{array}{c} \underset{d_{1},\ldots ,d_{N-1}}{\mathrm{argmin}}\;\;\left\{\underset{u_{s}\in U_{opt}}{\max}\;\;\zeta \left(SNR_{0},d_{1},\;\ldots ,\;d_{N-1},u_{s}\right)\right\}\;\\ s.t.\;\;P_{m}\left(SNR_{0},d_{1},\;\ldots ,\;d_{N-1},u_{s}\right)<P_{max},\;\;u_{m}\in U_{int},\;\;u_{s}\in U_{max}\; \end{array}$$

Since this array optimization approach is based on achieving the minimum MSE while keeping the probability of outliers under control, the obtained solution is referred to as the best outlier controlled array (BOCA). Its implementation diagram is sketched in Figure 7. The four blocks perform the following steps:

(i)Identify the APP region in the plane $\left(\Delta_{1},\Delta_{2}\right)$, using the procedure in Section 3;(ii)Set the value of μ, the maximum value for $u-u_{s}$, based on the sensor estimation characteristic and operational scenario;(iii)Map the APP region into the $\left(d_{1},d_{2}\right)$ plane, based on the value of μ;(iv)Look for the array configuration $\left(d_{1},d_{2}\right)$ providing the minimum of the MSE inside this region.

Notice that the APP region resulting from step (i) has been evaluated in the $\left(\Delta_{1},\Delta_{2}\right)$ plane. Recalling the definition of $\Delta_{n}$ in Equation (12), the APP region in Figure 6 can be remapped in the $\left(d_{1},d_{2}\right)$ plane by scaling the maximum acceptable value $\Delta_{1}^{max}\left(\alpha \right)$ for every slope α by $\lambda /\mu_{max}$. Hence, the inter-element distances $d_{n}$ of the array configurations inside the APP region are given by
(27)$$d_{n}\left(\alpha \right)<d_{1}{}_{max}\left(\alpha \right)=\frac{\lambda \Delta_{1}{}_{max}\left(\alpha \right)}{\mu_{max}}.$$

The scaling in Equation (27) requires knowledge of $\mu_{max}$, which in turn requires defining the sectors $U_{max},$ and $U_{int}$. The bounds of these sectors largely depend on the considered application and purpose, as illustrated in the following section.

Equation (27) allows us to represent the APP region in the array inter-element distances axis system, i.e., in the $\left(d_{1},d_{2}\right)$ plane, thus fulfilling step (iii). The final optimization algorithm (step (iv)) is based on a minimax criterion by choosing the array configuration that minimizes the maximum MSE obtained inside a given angular sector $U_{opt}=\left[-\sin\left(\theta_{opt}\right),\sin\left(\theta_{opt}\right)\right]$.

When compared to conventional, well-known array design strategies, such as the one presented in [28], the proposed approach offers three main advantages:

It allows for the relaxation of the half-wavelength quantization constraint on the inter-element distance.It offers a high level of flexibility, as it allows us to distinguish the angular sector $U_{max}$ where the source can be located, the angular sector $U_{int}$ where the source is pursued, and the angular sector $U_{opt}$, where the performance is optimized.It allows us to obtain satisfactory DoA estimation performances, even when the SNR is in the threshold region, thanks to the constraint imposed on the maximum pairwise error probability.

In order to illustrate these advantages, in the next section, we assess the performances obtained by the BOCA in different case studies via numerical analysis.

## 5. Performance Assessment on Simulated Data

The purpose of this section is to test the proposed design approach against simulated data in order to prove its effectiveness. In addition, we compare it to the conventional design strategy presented in [28], where the array inter-element distances are subject to a λ/2 quantization. To make a fair comparison, we compare the proposed BOCA with a quantized array that satisfies the same constraint on the maximum pairwise probability. This can be achieved by selecting one of the quantized array configurations falling inside the APP region, as conducted for the BOCA. The selected array will be referred to as λ/2-best outlier controlled array (λ/2-BOCA). Due to the quantization constraint, we expect that the λ/2-BOCA will obtain an equal or higher MSE than the BOCA. Furthermore, there might be cases in which a λ/2-BOCA is impossible to locate, as all quantized configurations fall outside the APP region.

As mentioned in the previous sections, the APP region depends on both $\theta_{max}$ and $\theta_{int}$, while the selected BOCA inside the APP region changes with $\theta_{opt}$. Since different values for $\theta_{max}$, $\theta_{int}$, and $\theta_{opt}$ can be selected based on the requirements of different applications, we define two case studies that will be used for performance assessment: 

(i)Case A: $\theta_{max}=90{}^{\circ}$, so that the source can be located at any possible DoA. Therefore, the possibility of having sources outside the angular sector $U_{int}$ of interest is considered.(ii)Case B: $\theta_{max}=\theta_{int}$, so that the possibility of having targets outside a given angular sector $\left[-u_{max},u_{max}\right]$ is excluded. This can be useful when electromagnetic shielding is employed, or when the antennas are characterized by a very directive pattern.

Both case studies are considered in Section 5 and Section 6, respectively dedicated to unconstrained and constrained design solutions. Notice that restricting $\theta_{max}$, as in case B, leads to smaller values of μ, and thus to larger APP regions, as shown in Equation (27). In turn, this allows us to choose longer array configurations, which exhibit better performances in terms of MSE.

To study the performance of the resulting BOCA, for both Case A and Case B, we evaluate the MSE of its DoA estimate inside the area of interest ($-\theta_{opt}$, $\theta_{opt}$) as a function of the two remaining parameters, namely $\theta_{opt}$ and $\theta_{int}$, assuming assigned values for SNR and $P_{max}$. Since the MSE is variable in this region, a compact method to represent the performance is to average its value inside the sector $U_{opt}$. This provides us with a single averaged MSE value for each choice of $\theta_{opt}$ and $\theta_{int},$ since $\theta_{opt}\leq \theta_{int}$, the angular sector $U_{opt}$ in which performances are studied, is a subset of the sector $U_{int}$ of the sources of interest, i.e.,: $U_{opt}\subseteq U_{int}$.

To show numerical results, we assume an SNR value of 20 dB, and we fix $P_{max}=10^{-4}$. Figure 8a,b shows the map of the average MSE for case A and case B, respectively, for all the admissible values of $\theta_{int}$ and $\theta_{opt}$. According to these results, the following observations apply:

(i)Wider angular sectors $U_{opt}$ result in higher values of MSE, whereas when we are interested only to the DoA estimate inside a narrow angular sector, much better MSE values can be obtained. (ii)For greater values of $\theta_{int}$ the admissible region becomes smaller, leading to BOCA solutions with a higher MSE.(iii)Hence, the larger the $\theta_{int}$ and $\theta_{opt}$ are, the higher is the average MSE.(iv)Case B provides generally lower values for the average MSE, since the smaller value of $\theta_{max}$ provides a wider APP region, thus increasing the opportunity to select a longer BOCA configuration.

To compare the performance of the BOCA and the λ/2-BOCA, it is convenient to define the ratio $R\left(\theta_{opt},\theta_{int}\right)$ between the two mean MSEs, namely:(28)$$R\left(\theta_{opt},\theta_{int}\right)=\frac{E\left\{MSE_{\lambda /2-BOCA}\left(\theta_{opt},\theta_{int}\right)\right\}}{E\left\{MSE_{BOCA}\left(\theta_{opt},\theta_{int}\right)\right\}}.$$

While the maximum (worst) MSE inside the sector $U_{opt}$ for the BOCA is always smaller compared to the λ/2-BOCA, the ratio $R\left(\theta_{opt},\theta_{int}\right)$ of the averaged MSEs over the sector $U_{opt}$ are not necessarily greater than the unity. This ratio allows us to quantify the average improvement achieved by the proposed approach over the conventional method, as a function of $\theta_{opt}$ and $\theta_{int}$, over the whole area of interest. Figure 9a,b shows the $R\left(\theta_{opt},\theta_{int}\right)$ map for case study A and case study B, respectively.

By observing these results, the following remarks apply:

(i)The maximum achievable ratio is $R\left(\theta_{opt},\theta_{int}\right)≅3$. This means that $E\left\{MSE_{\lambda /2-BOCA}\left(\theta_{opt},\theta_{int}\right)\right\}$ is at most about 3.5 times higher than $E\left\{MSE_{BOCA}\left(\theta_{opt},\theta_{int}\right)\right\}$).(ii)As anticipated at the beginning of this section, there are cases in which no λ/2-BOCA can be found. For the selected parameters SNR=20 dB and $P_{max}=10^{-4}$, this occurs when choosing $\theta_{a}\geq 80{}^{\circ}$. In these cases, the ratio shown in the figures has been conveniently saturated to its maximum value, namely 3.5, to denote that the improvement is not measurable.(iii)As previously mentioned, in case B, we obtain wider APP regions. Therefore, longer arrays are generally admissible, and the average MSE in case B is always smaller or than or equal to the average MSE in case A, for any given pair of $\theta_{int}$ and $\theta_{opt}$.(iv)Finally, while in case A, the best ratio values are obtained for smaller $\theta_{int}$ and $\theta_{opt}$, in case B, the best ratio values are obtained for $\theta_{max}=\theta_{int}=\theta_{opt}\approx 30{}^{\circ}$.

To complete the analysis, we show the detailed performance analysis for few specific parameter sets $\left(\theta_{opt},\theta_{int}\right)$*,* corresponding to the points of the $R\left(\theta_{opt},\theta_{int}\right)$ maps in Figure 9a,b, as shown with black circles, with the selected parameters $\theta_{max},\;\theta_{int}$, and $\theta_{opt}$ for each case study.

### 5.1. Case Study A—Source Located within the Whole [−90°,90°] Angular Sector ($\theta_{max}=90{}^{\circ}$)

In case study A, we focus on three subcases characterized by $\theta_{int}=30{}^{\circ}$. However, while the average MSE of both the BOCA and the λ/2-BOCA monotonically increase with $\theta_{opt}$, their ratio shows a different behavior, namely $R\left(\theta_{opt},\theta_{int}\right)\approx 1.25$ at $\theta_{opt}=18{}^{\circ}$ and $R\left(\theta_{opt},\theta_{int}\right)\approx 1.7$ at both $\theta_{opt}=6{}^{\circ}$ and $\theta_{opt}=30{}^{\circ}$. As the considered subcases are characterized by the same values of $\theta_{max},\theta_{int},SNR,$ and $P_{max}$, they all share the same admissible region, as represented in Figure 10.

The location of the BOCA and λ/2-BOCA for the three subcases is represented with a blue and a green circle, respectively, in the three subplots. With the selected set of parameters, the optimization procedure leads to the BOCA and λ/2-BOCA configurations reported in Table 2.

Figure 11a–c shows the MSE obtained using the $z_{BOCA}$ and the $z_{\lambda /2-BOCA}$, respectively, for case studies A1, A2, and A3 as a function of θ. The solid curves represent the theoretical MSE, while the dashed examples represent CRB. Furthermore, dots represent the MSE obtained through simulation, and the vertical dash-dot lines denote the area where we look for the optimal solution.

The numerical MSE values were obtained through a Monte Carlo simulation, with $N_{MC}=10^{5}$ runs. The simulated array output x was generated according to the signal model in Equation (2), where the N random samples in the noise vector n were assumed to be complex valued and Gaussian distributed, namely: $n\sim CN\left(0,\sigma_{n}^{2}\right)$. The complex-valued amplitude A of the source baseband signal is a deterministic parameter, set so as to guarantee the required SNR condition, namely $A=\sqrt{2\sigma_{n}^{2}SNR}$.

Figure 11a shows the MSE of $z_{BOCA}=\left[0.00\;1.30\;5.84\right]$ to be lower than those of $z_{\lambda /2-BOCA}$= [0.00, 1.00, 4.50] (see × markers in Figure 10) for $\left|\theta_{s}\right|\leq 6{}^{\circ}$; however, for $\left|\theta_{s}\right|\geq 12{}^{\circ}$, the BOCA is subject to the presence of a sidelobe, and its MSE shows a step increase, whereas the λ/2-BOCA is not subject to this effect. As clear from Figure 11b, for values of $\left|\theta_{s}\right|\geq 12{}^{\circ}$, the BOCA configuration changes to $z_{BOCA}=\left[0.00\;0.66\;5.22\right]$, while the λ/2-BOCA remains unchanged (see ○ markers in Figure 10). 

While the new BOCA still has a lower MSE than the λ/2-BOCA, its value is not as low as it was previously. When $\left|\theta_{s}\right|\geq 22{}^{\circ}$, the array [0.00, 1.00, 4.50] is subject to the presence of a sidelobe, which provides a step increase in the MSE, so that the λ/2-BOCA configuration changes to $z_{\lambda /2-BOCA}=\left[0.00,\;0.50,\;4.00\right]$, as apparent from Figure 11c. Its average MSE value increases, whereas the BOCA configuration is not subject to significant changes (see + markers in Figure 10).

We also observe that when the MSE is too high, the simulated and theoretical MSE curves do not match exactly. This is because the union-bound approximation in Equations (7) and (8) is not tight.

To complete the analysis, Figure 12 shows the maximum pairwise probability for each array configuration. The vertical dash-dot lines denote the area where the constraint must be guaranteed, while the horizontal dashed line represents the constraint value $P_{max}$. Figure 12 illustrates that both the λ/2-BOCA and the BOCA satisfy the constraint on the maximum pairwise probability inside the angular sector $U_{int}$, as we expected. However, as shown in Figure 11, the BOCA performs better inside the $\left[-\theta_{opt},\theta_{opt}\right]$ angular sector in terms of MSE.

### 5.2. Case Study B—Source Only Located within the Angular Sector $\left[-\theta_{int},\theta_{int}\right]$ of the DoA of Interest $\left(\theta_{max}=\theta_{int}\right)$

As opposed to case study A, in case B, the possibility that a source might be located outside the angular sector of interest is excluded, i.e., $\theta_{max}=\theta_{int}$. We focus on two subcases characterized by $\theta_{int}=30{}^{\circ}$, as they seem to be the cases in which the best ratios $R\left(\theta_{opt},\theta_{int}\right)$ are obtained. As in case A, the different subcases are characterized by the same admissible region, as represented in Figure 13. With the selected set of parameters, the optimization procedure leads to the BOCA and λ/2-BOCA reported in Table 2. 

Note that the decrease in $\theta_{max}$ yields a larger admissible region. This allows us to choose longer array configurations, which were non-admissible in case A. Despite this, the λ/2-BOCA is still the one selected in the previous cases, confirming the reduced flexibility of the quantized approach. These considerations are further confirmed by Figure 14a,b and Figure 15a,b, where the MSE and the maximum pairwise error probability obtained using the different configurations is evaluated across a grid of θ values.

As a final consideration, we notice that the admissible regions shown in Figure 10 and Figure 13, have very spiky shapes in certain areas. Selecting an array configuration inside one of those thin portions would mean requiring an installation accuracy that would be unlikely to be guaranteed in practice. Furthermore, these types of solutions are highly unstable, and lead to the variability of the ratio $R\left(\theta_{opt},\theta_{int}\right)$ with $\theta_{opt}$. Therefore, in the next section, we modify the optimization problem by adding additional constraints that might be required for the practical application of the proposed approach.

## 6. Constrained Three-Element NULA Design Strategy

In this section, we consider two technological constraints that are likely to arise in real case scenarios, and we introduce a slight modification in the proposed strategy that will make it suitable for practical applications.

The first technological constraint is introduced based on the following observation: the APP regions shown in the previous sections always include solutions in which the inter-element distance is very small. In practical applications, depending on the signal wavelength, the physical size of the antennas, and the coupling effects that might occur, it might not be possible to position the antenna elements at the required proximity. Therefore, we consider the possibility to exclude the solutions characterized by inter-element distances shorter than a fixed value l from the admissible region.

The second technological constraint is related to installation accuracy. We define a finite installation accuracy δ, representing the maximum error tolerated in the installation. Specifically, a given array $\bar{z}=\left[0\;{\bar{d}}_{1}\;{\bar{d}}_{2}\right]$ inside the APP region is considered admissible only if the four arrays $\left[0\;\left({\bar{d}}_{1}\pm \delta \right)\;\left({\bar{d}}_{2}\pm \delta \right)\right]$ also lie within the admissible region. This excludes all the thinner areas from the admissible region.

The block diagram in Figure 16 illustrates the constrained optimization procedure. Clearly, for l=0 and δ=0, the constrained design strategy corresponds to the one in Equation (26).

To show the effect of the additional constraints on the results, let us consider a passive location system operating in the Wi-Fi band at $f_{0}=2.447\;\mathrm{GHz}$ (λ=0.1226 m). We assume that the employed commercial antennas have a size l≈0.13 m=1.1λ, constraining the minimum inter-element distance. Two different values are considered for δ, namely $\delta =0.005\lambda =6.1\cdot 10^{-4\;}m$ (micrometric positioning) and $\delta =0.05\lambda =6.1\cdot 10^{-3}\;m$ (manual positioning). Figure 17a,b shows the admissible region obtained by including the technological constraints for the two cases, assuming SNR=20 dB, $P_{max}=10^{-4}$ and $\theta_{max}=\theta_{int}=30{}^{\circ}$. The blue-colored areas are the portions of the APP region excluded by the constraint on the minimum inter-element distance, while the red-colored areas are the portions of the APP excluded to consider installation accuracy tolerance. As visible from Figure 17, the installation accuracy constraint removes thinner areas from the APP region. These areas contain array configurations with highly unstable performance, for which a slight modification in the inter-element distance causes the array configuration to fall outside the APP region.

As visible in Figure 17, the technological constraints lead to significant changes in the APP region, so that most of the array configurations considered in case studies A1-A3, and B1 and B2, are no longer admissible. Therefore, for each choice of $\theta_{max}$, $\theta_{int}$, and $\theta_{opt}$, new BOCA and λ/2-BOCA configurations must be identified. For example, considering l=1.1λ and δ=0.05λ, as shown in Figure 17b, and assuming $\theta_{max}=\theta_{int}=\theta_{opt}=30{}^{\circ}\;$, the new BOCA configurations become $z_{BOCA}=\left[0\;1.96\;6.82\right]\lambda$ and $z_{\lambda /2-BOCA}=\left[0\;1.5\;3.5\right]\lambda$. We also notice that the obtained BOCA configuration $z_{BOCA}$ is characterized by a very low value of $min\left(S_{pq}\right)=4.74$, as evaluated following the procedure in [31], and it would have not been considered admissible based on the maximum error criterion. However, $z_{BOCA}$ is guaranteed to satisfy the constraint $P_{m}<P_{max}$, since it belongs to the APP region. Therefore, in this case, the $min\left(S_{pq}\right)$ value is not a good proxy for robustness to estimation ambiguities, since our ML-based admissibility criterion allows us to find a robust array configuration with low $min\left(S_{pq}\right)$. This is further confirmed by observing the ratio $R\left(\theta_{opt},\theta_{int}\right)$ between the MSEs of the $z_{BOCA}$ and that of the $z_{\lambda /2-BOCA}$. Since we have $R\left(\theta_{opt},\theta_{int}\right)=3.1510$, the BOCA design solution ultimately achieves an MSE which is about one-third of the MSE obtained by the λ/2-BOCA. In conclusion, $z_{BOCA}$ is indeed a good alternative to both the λ/2-BOCA and to the BOCA designs obtained in the case studies without technological constraints. 

Depending on the value of the constraints and on the extrema of the considered angular sectors, the modified APP region might become so restrictive that no λ/2-BOCA configurations could be found. In such cases, the ratio $R\left(\theta_{opt},\theta_{int}\right)$ is saturated at its maximum value, namely $R\left(\theta_{opt},\theta_{int}\right)=3.5$. Figure 18a,b shows the contour plots of $R\left(\theta_{opt},\theta_{int}\right)$, obtained considering case studies A and B, respectively, and assuming an installation accuracy tolerance of δ=0.005λ. As is apparent, the additional design constraints degrade the estimation accuracies of both the λ/2-BOCA and the BOCA configurations. Therefore, the ratio R between the two MSEs still assumes values between R≅1 and R≅3.5.

By increasing the installation accuracy tolerance to δ=0.05λ, we obtain the contour plots of $R\left(\theta_{opt},\theta_{int}\right)$, as shown in Figure 19a,b, relative to cases A and B, respectively.

As visible from these contour plots, the effect of the higher tolerance can provide a significant change in the array design. Specifically, most of the $\left(\theta_{opt},\theta_{int}\right)$ plane is saturated to 3.5. Based on this observation, we conclude that the proposed approach is always able to provide a valid array configuration, while there is no λ/2-BOCA solution that satisfies all the imposed constraints at the same time.

In conclusion, the results obtained in case study B, assuming δ=0.005λ, are also used to validate the obtained design strategy in a real-case scenario using experimental data, as reported in the following section.

## 7. Experimental Results

To further support the effectiveness of the design procedure proposed in the previous sections, an experimental trial was conducted in a parking area.

For this experiment, we assumed $\theta_{max}=\theta_{int}=45{}^{\circ}$. The design of the array has been carried out to guarantee $P_{m}<P_{max}=10^{-4}$ when the operative SNR value is set to $SNR_{0}=20\;\mathrm{dB}$. Finally, the minimum inter-element distance has been constrained to be l=1.1λ, as in the previous section, with δ=0.005λ. With these parameters, we obtain the admissible region in Figure 20.

To carry out the experimental acquisition, we exploited a four-channel National Instruments NI USRP-2955 board, operating in the 2.5 GHz Wi-Fi frequency band. To collect samples simultaneously with the two arrays, we must operate with two configurations that have two elements in common. For this purpose, we selected a non-optimal array, addressed in the following as Almost-BOCA array $z_{ABOCA}=\left[0\;2\;5.35\right]\lambda$, that has one inter-element distance in common with the λ/2-BOCA, $z_{\lambda /2-\mathrm{BOCA}}=\left[0\;2\;4.5\right]$. This allowed us to make the acquisitions with a single 4-elements array with z=[0 2 4.5 5.35], with the individual element connected channels 0,1,2,3 of the 4-channel USRP acting as the coherent receiver. With this arrangement, the snapshots of the λ/2-BOCA array were obtained by using only the samples received by channels 0,1, and 2, whereas those of the Almost-BOCA array were obtained by using only the samples received by channels 0,1, and 3. Moreover, in this way, the snapshots of the two arrays are coherent and simultaneous. Note that the resulting array $z_{ABOCA}$ provides only slightly degraded performance with respect to the BOCA.

Figure 21a shows a sketch of the acquisition geometry. The green markers represent the element of the λ/2-BOCA, while the blue markers indicate the elements of the Almost-BOCA. The two sensors in the middle are common between the two arrays. The four-element array that includes both the Almost-BOCA and the λ/2-BOCA solutions was realized by deploying four Wi-Fi antennas on a plastic support mounted on a tripod. Figure 21b shows this receiving system set in the origin of the reference system. The Wi-Fi access point (AP) in Figure 21c is used as the transmitting source, which has been initially placed at 20 m with DoA $\theta_{s}=0{}^{\circ}$ from the receiving antennas and then moved to $\theta_{s}=30{}^{\circ}$ with respect to the boresight of the array. This was set to transmit the Wi-Fi beacons with a beacon interval of 3ms at $f_{0}=2.447\;\mathrm{GHz}$, (namely, using wavelength λ=0.1226 m). 

The expected performance for SNR=20 dB in terms of both MSE and $P_{out}$ is reported in Figure 22a,b, as obtained from the theoretical framework in Equations (7)–(9) for both the Almost-BOCA and the λ/2-BOCA. The following remarks apply:

When the source is located at $\theta_{s}=0{}^{\circ}$, the MSE obtained with the λ/2-BOCA should be about 2 times higher than the MSE obtained with the Almost-BOCA, while the $P_{out}$ is about 4 times lower.

When the source is located at $\theta_{s}=30{}^{\circ}$, the MSE obtained with the λ/2-BOCA should be about 1.5 times higher than the MSE obtained with the Almost-BOCA, while the $P_{out}$ is about 3 times lower.

The experimental results have been obtained by simultaneously collecting long sequences of samples with the 4 elements when the AP is active. These data are received with a measured SNR=23 dB. White Gaussian noise was added to the data to emulate lower SNR values, specifically between 14 dB and 23 dB, with a 0.5 dB step.

By averaging the received samples, the resulting MSE and $P_{out}$ curves are obtained and reported in Figure 23a,b, respectively, as a function of the SNR value and for both considered DoAs. By observing Figure 23, we note that:

For both the considered DoAs, the Almost-BOCA solution outperforms the λ/2-BOCA in the asymptotic region, since a longer array is used;For $\theta_{s}=0{}^{\circ}$, the Almost-BOCA solution outperforms the λ/2-BOCA, both in terms of MSE and $P_{out}$ for SNR≥16 dB;For $\theta_{s}=30{}^{\circ}$, the improvement is lower as predicted by theory, and starts at a higher SNR (SNR≥19 dB).

As apparent, the experimental results show that the non-λ/2-spaced array always outperforms its λ/2 counterpart for the design value $SNR_{0}=20\;\mathrm{dB}$ and for higher *SNR* values. This confirms that the design procedure effectively allows us to design an array that provides a better performance than the λ/2-BOCA in a realistic experimental scenario.

We recall that a slightly different and better performing BOCA solution would be selected if we did not force it to have one inter-element distance in common with the λ/2-BOCA. Therefore, higher improvements are expected if the design strategy is applied with its entire set of degrees of freedom.

## 8. Conclusions

DoA estimation of narrowband signals is a ubiquitous task in several civil surveillance applications, such as passive coherent location systems, passive sonar, or passive radar. The main challenge in these applications is to comply with the typical low-cost requirements involving not only the economic cost, but also the computational load. Additionally, the DoA estimation accuracy is usually degraded by the presence of outliers, which occur as a result of the poor SNR conditions characterizing low-cost systems. To achieve the required features of limited complexity and system lightness, the number of antenna elements must be kept sufficiently low. With this perspective, NULA configurations represent a viable solution to reduce the number of receiving elements without compromising performance.

The design strategy presented in this paper is set within the context of low-cost sensors applications, as it allows us to identify the so-called best outlier-controlled NULA configuration, achieving satisfactory DoA estimation performance by exploiting just three antenna elements. As a matter of fact, the BOCA solution allows for minimizing the MSE, while keeping the outlier probability under reasonable control.

The design procedure is based on the following two steps. First, an admissible pairwise probability region is identified, exploiting the algorithm presented in Section 3. The APP region contains all the arrays for which the outlier probability is kept under control. Second, the BOCA configuration achieving the minimum MSE is located inside the APP region. It has also been shown that the APP region can be easily modified to incorporate practical design constraints, such as the minimum inter-element distance and the installation accuracy tolerance.

The proposed design strategy has been compared to the one presented in [28], which represents our benchmark. Therein, the array inter-element distances are quantized to half-wavelength. To carry out a fair analysis, we compared the estimation accuracy achieved by the BOCA with the one achieved by the best outlier-controlled array with the half-wavelength quantization constraint, namely the λ/2-BOCA. As shown via both theoretical and numerical analyses, the BOCA solutions generally achieve improved estimation accuracies compared to the λ/2-BOCA estimations, with an improvement depending on how the angular sectors $U_{max}$, $U_{int}$, and $U_{opt}$ are chosen. The effectiveness of the proposed design procedure has also been verified using experimental data, confirming that the optimized array outperforms the λ/2-quantized benchmark array. The comparison with a recent three-element NULA design approach to control ambiguity outliers is also included, which emphasizes the capability of our proposed model to accurately characterize and control the probability of ambiguities.

As a future research scope, it might be interesting to extend the design strategy to four-element and five-element NULA configurations in order to improve the estimation accuracy without significantly increasing the number of receiving channels. In such cases, alternative design strategies, such as nested or coprime array configurations, may also provide a useful benchmark. Furthermore, non-linear array configurations, such as circular arrays, could be investigated, as they would allow us to estimate the DoA regarding both azimuth and elevation. Lastly, the DoA estimation problem could be extended to multi-source scenarios, which would be useful not only when multiple sources of interest are present, but also to take any multipath effect into proper account.

## Figures and Tables

**Figure 1 sensors-23-04872-f001:**
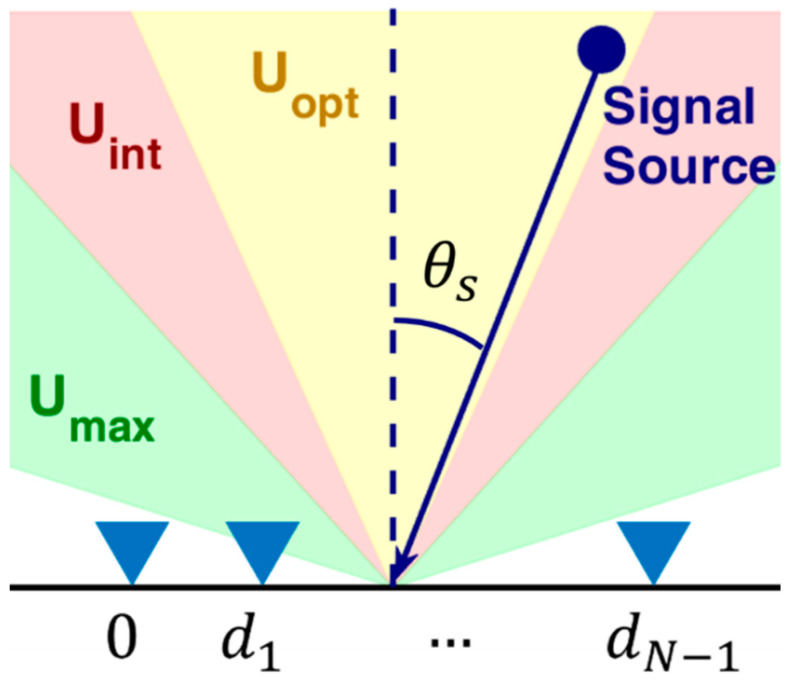
Array geometry, operative scenario, and angular sectors of interest.

**Figure 2 sensors-23-04872-f002:**
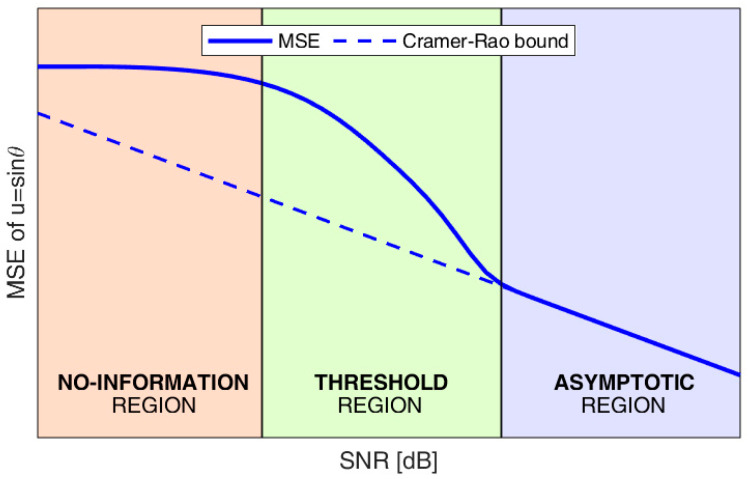
Qualitative behavior of the mean square error (MSE) vs. signal-to-noise power ratio (SNR).

**Figure 3 sensors-23-04872-f003:**
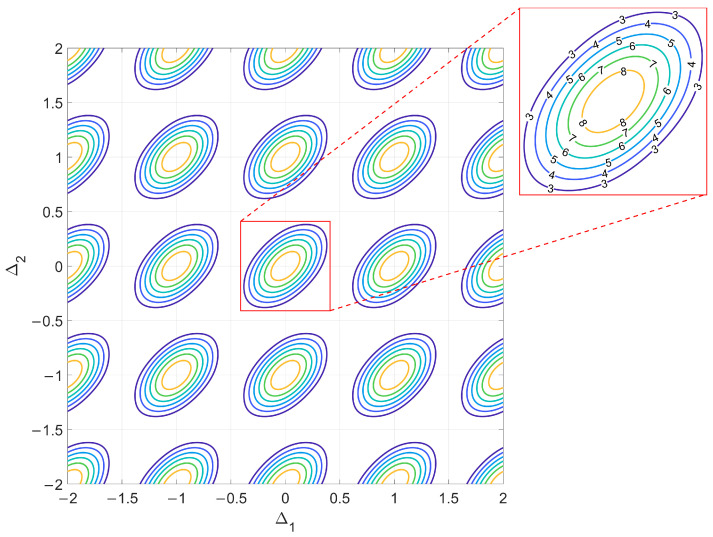
Contour plot of the ideal likelihood function $L\left(\Delta_{1},\Delta_{2}\right)$.

**Figure 4 sensors-23-04872-f004:**
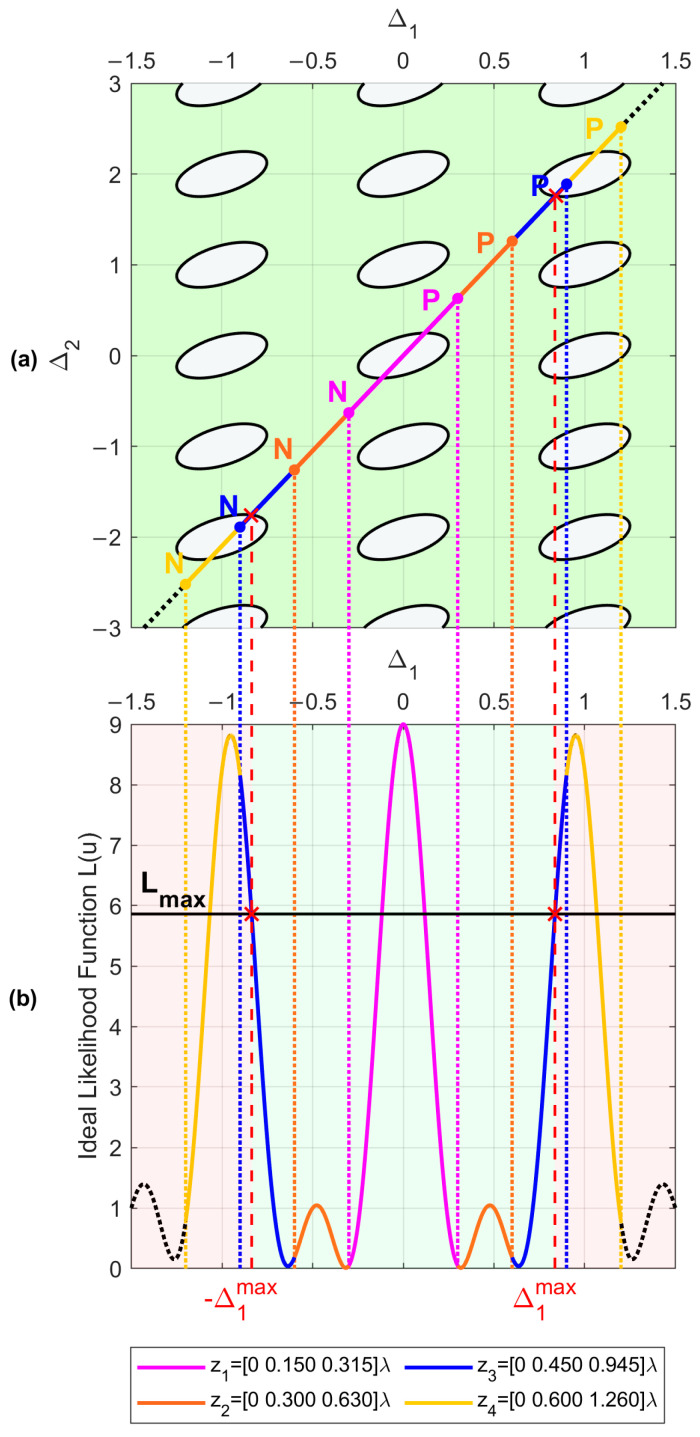
(**a**) Representation of the considered array configurations as segments on the plane $\left(\Delta_{1},\Delta_{2}\right).$ (**b**) Beampattern of the considered array configurations and array admissibility.

**Figure 5 sensors-23-04872-f005:**
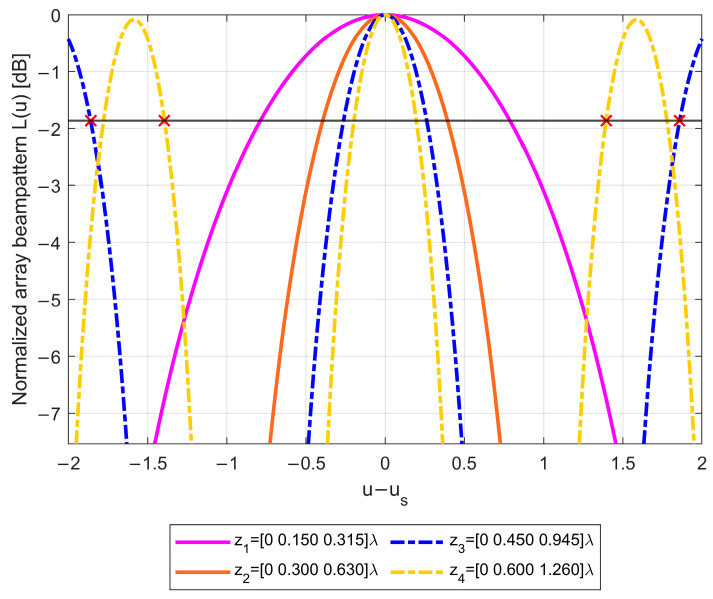
Normalized beampatterns of the four array configurations.

**Figure 6 sensors-23-04872-f006:**
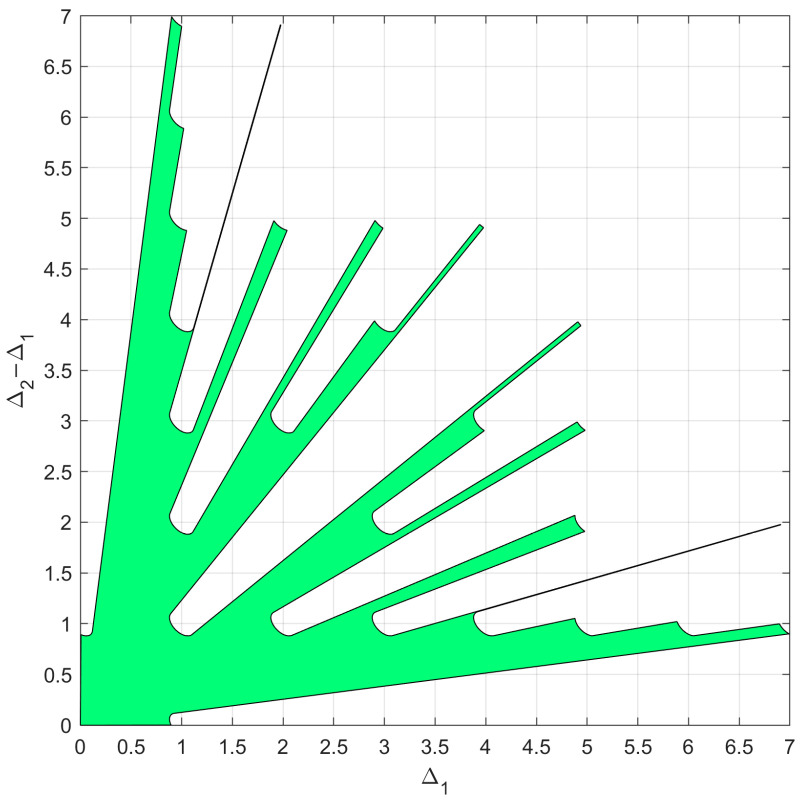
Admissible region in the plane $\left(\Delta_{1},\Delta_{2}-\Delta_{1}\right)$ obtained for the case of SNR=20dB and $P_{max}=10^{-4}$.

**Figure 7 sensors-23-04872-f007:**
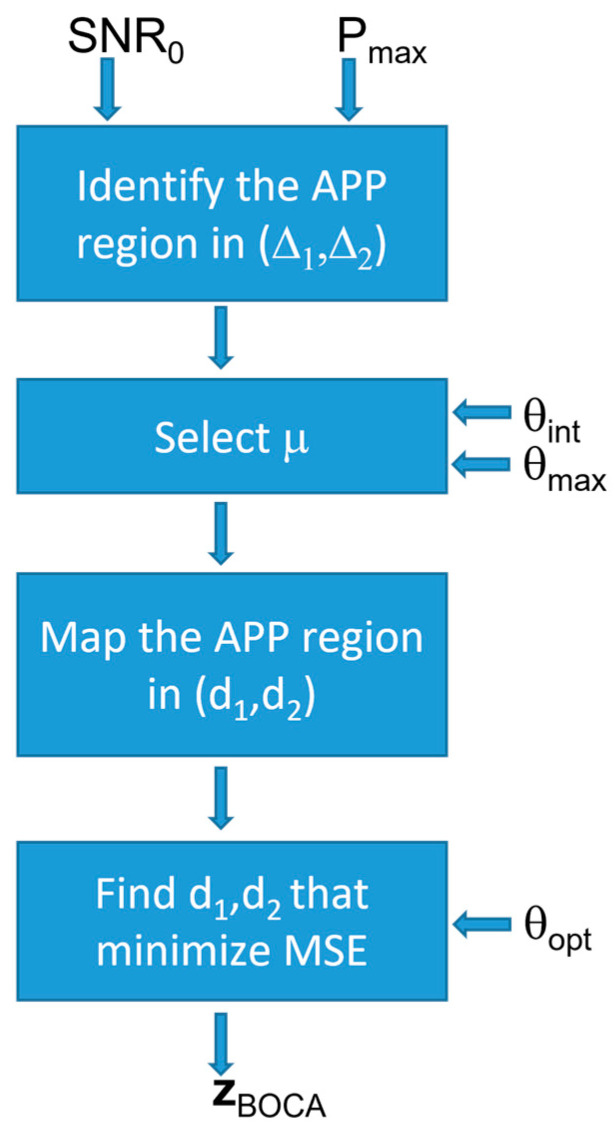
Block diagram illustrating the best outlier-controlled array (BOCA) design strategy.

**Figure 8 sensors-23-04872-f008:**
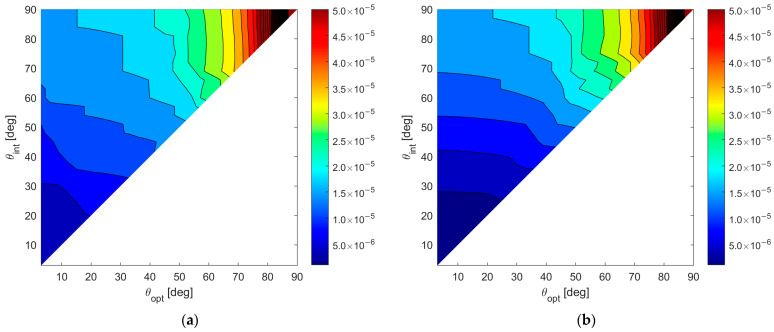
A 2D contour plot of the MSE averaged inside $U_{opt}=\left[-sin\theta_{opt},sin\theta_{opt}\right]$ as a function of $\theta_{opt}$ and $\theta_{int}$; (**a**) case study A ($\theta_{max}=90{}^{\circ}$); (**b**) case study B ($\theta_{max}=\theta_{int}$).

**Figure 9 sensors-23-04872-f009:**
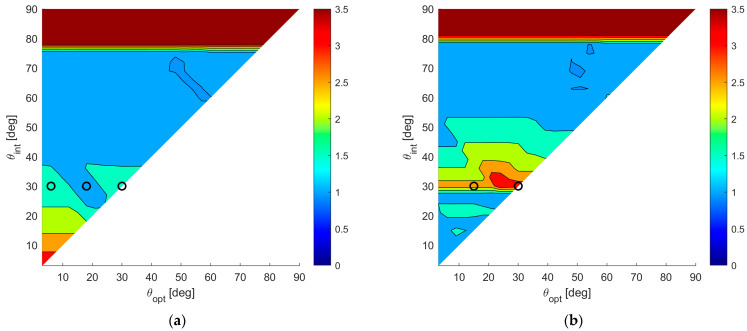
A 2D contour plot of the ratio $R\left(\theta_{opt},\theta_{int}\right)$ as a function of $\theta_{opt}$ and $\theta_{int}$; (**a**) case study A ($\theta_{max}=90{}^{\circ}$); (**b**) case study B ($\theta_{max}=\theta_{int}$).

**Figure 10 sensors-23-04872-f010:**
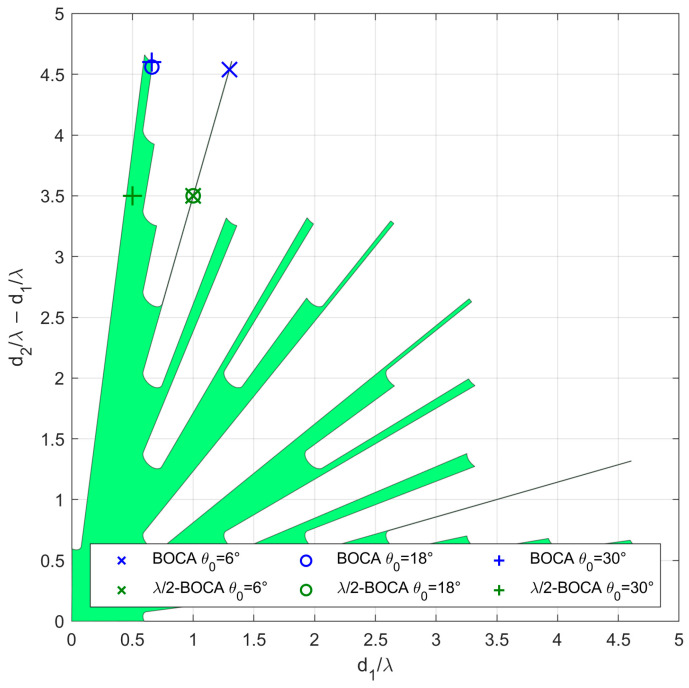
Admissible region and obtained solutions for case study A with $\theta_{int}=30{}^{\circ}$, SNR=20dB, $P_{max}=10^{-4}$, and (×) $\theta_{opt}=6{}^{\circ}$, (o) $\theta_{opt}=18{}^{\circ}$, (+) $\theta_{opt}=30{}^{\circ}$.

**Figure 11 sensors-23-04872-f011:**
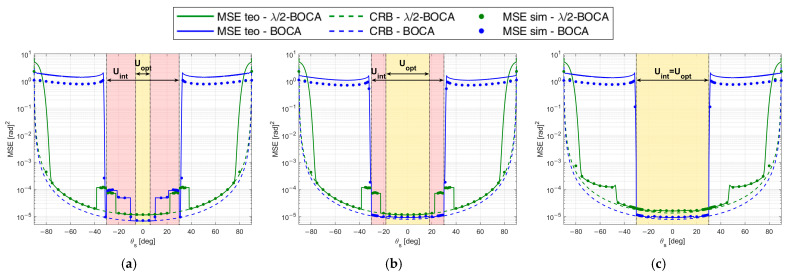
MSE of BOCA and λ/2-BOCA for case study A: $\theta_{int}=30{}^{\circ}$, SNR=20 dB, $P_{max}=10^{-4}$, and (**a**) $\theta_{opt}=6{}^{\circ}$; (**b**) $\theta_{opt}=18{}^{\circ}$; (**c**) $\theta_{opt}=30{}^{\circ}$.

**Figure 12 sensors-23-04872-f012:**
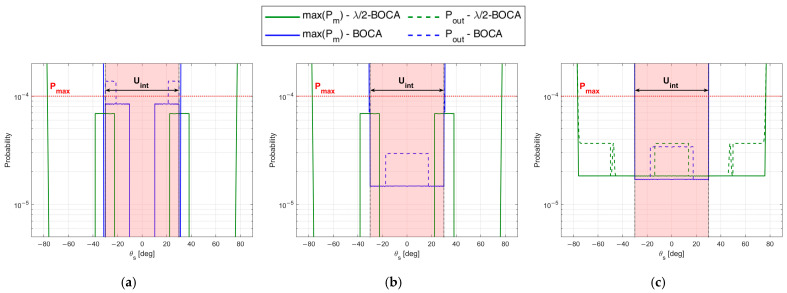
Maximum pairwise error probability of BOCA and λ/2-BOCA for case study A: $\theta_{int}=30{}^{\circ}$, SNR=20 dB, $P_{max}=10^{-4},\;$ and (**a**) $\theta_{opt}=6{}^{\circ}$, (**b**) $\theta_{opt}=18{}^{\circ}$, (**c**) $\theta_{opt}=30{}^{\circ}$.

**Figure 13 sensors-23-04872-f013:**
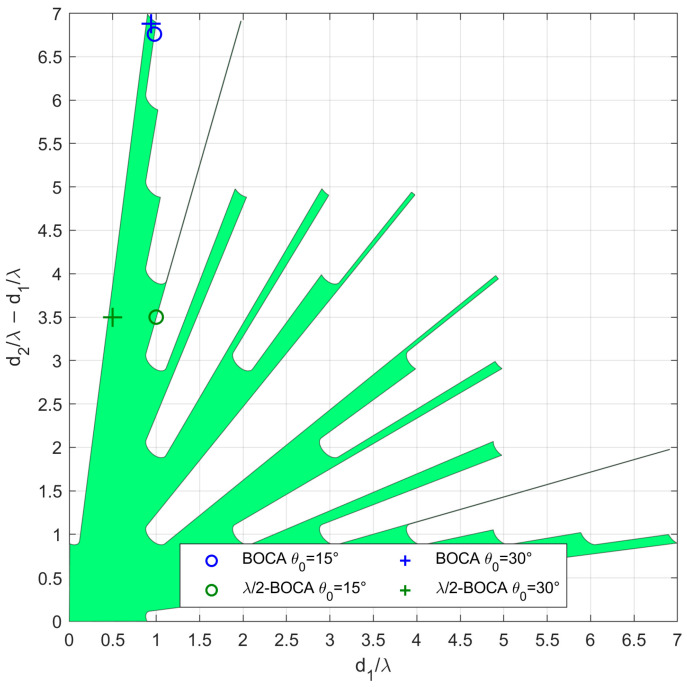
Admissible region and obtained solutions for case study B with $\theta_{int}=30{}^{\circ}$, SNR=20dB, $P_{max}=10^{-4}$, and (o) $\theta_{opt}=15{}^{\circ}$, (+) $\theta_{opt}=30{}^{\circ}$.

**Figure 14 sensors-23-04872-f014:**
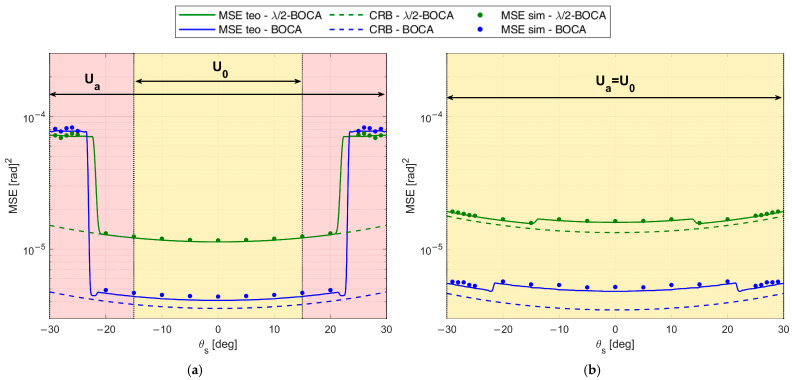
MSE of BOCA and *λ*/2-BOCA for case study B: $\theta_{int}=30{}^{\circ}$, SNR=20 dB, $P_{max}=10^{-4}$, and (**a**) $\theta_{opt}=6{}^{\circ}$, (**b**) $\theta_{opt}=30{}^{\circ}$.

**Figure 15 sensors-23-04872-f015:**
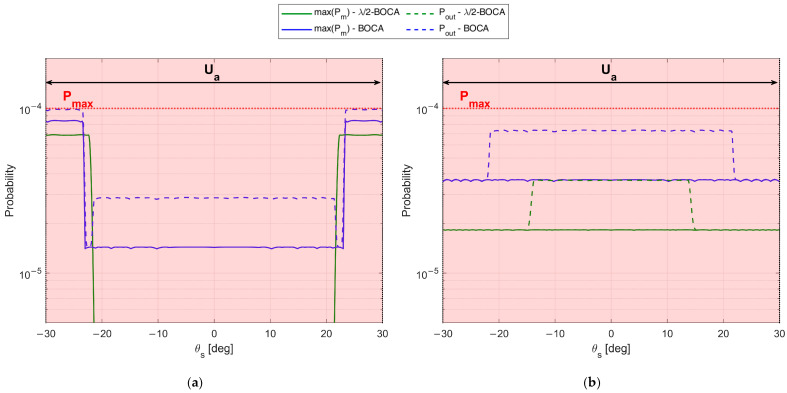
Maximum pairwise error probability of BOCA and λ/2-BOCA for case study B: $\theta_{int}=30{}^{\circ}$, SNR=20 dB, $P_{max}=10^{-4}$, and (**a**) $\theta_{opt}=6{}^{\circ}$, (**b**) $\theta_{opt}=30{}^{\circ}$.

**Figure 16 sensors-23-04872-f016:**
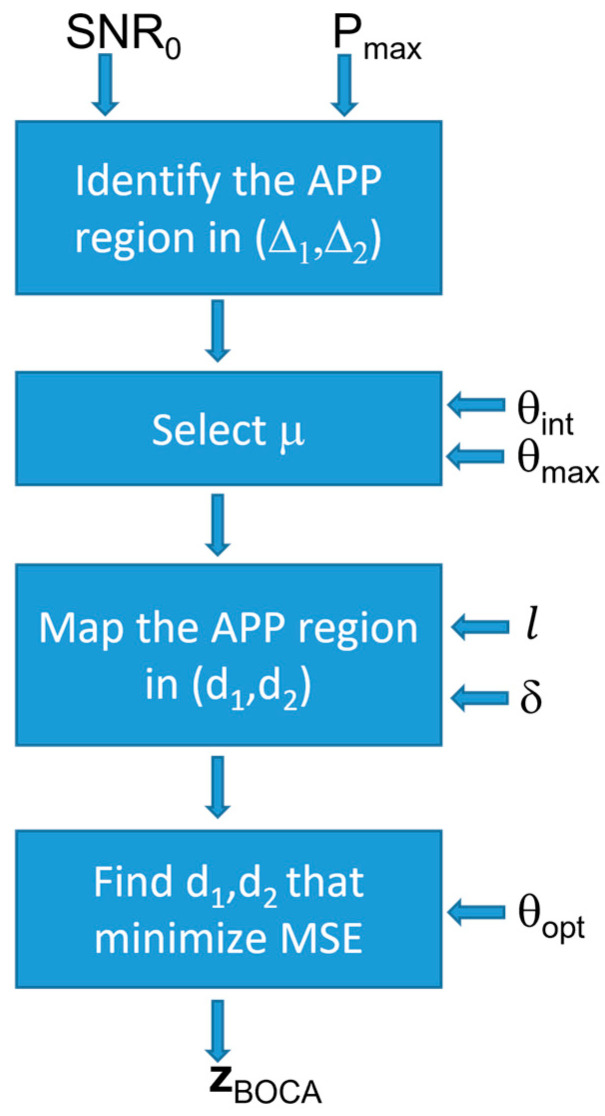
Block diagram illustrating the constrained design strategy.

**Figure 17 sensors-23-04872-f017:**
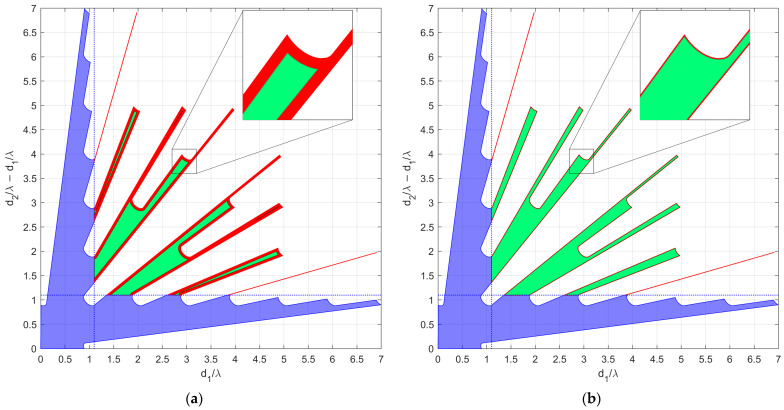
Admissible region for the constrained case, with $\theta_{max}=\theta_{int}=30{}^{\circ}$, SNR=20 dB, $P_{max}=10^{-4}$, l=1.1λ, and (**a**) δ=0.005λ, (**b**) δ=0.05λ.

**Figure 18 sensors-23-04872-f018:**
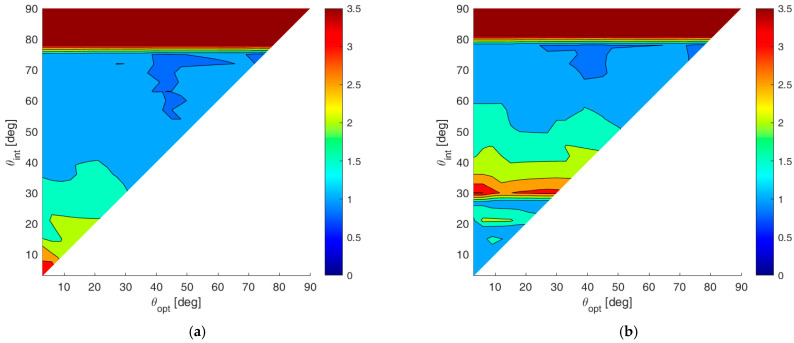
2D improvement maps for the constrained case with δ=0.005λ: (**a**) case study A; (**b**) case study B.

**Figure 19 sensors-23-04872-f019:**
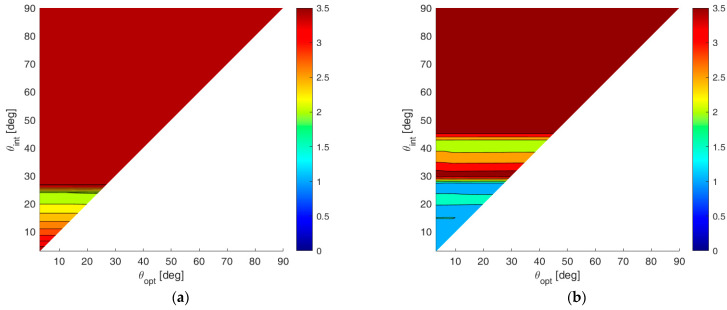
The 2D improvement maps for the constrained case with δ=0.05λ: (**a**) case study A; (**b**) case study B.

**Figure 20 sensors-23-04872-f020:**
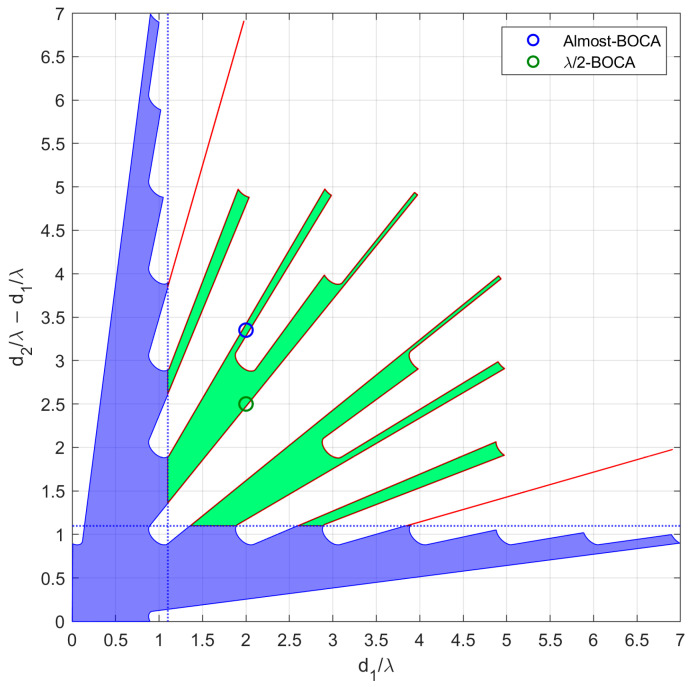
Admissible region obtained for SNR=20 dB and $P_{out}=10^{-4}$. The green dot represents the λ/2-BOCA, while the blue dot is the Almost-BOCA, modified with the purpose of having one inter-element distance in common with the λ/2-BOCA.

**Figure 21 sensors-23-04872-f021:**
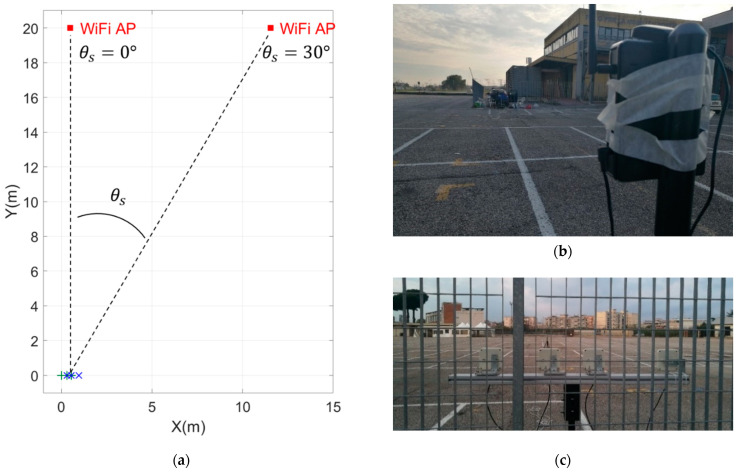
Acquisition geometry: (**a**) sketch of the transmitter and receiver configurations (**b**) Wi-Fi access point transmitter to be located, and (**c**) receiving array geometry with N=4 Wi-Fi antennas.

**Figure 22 sensors-23-04872-f022:**
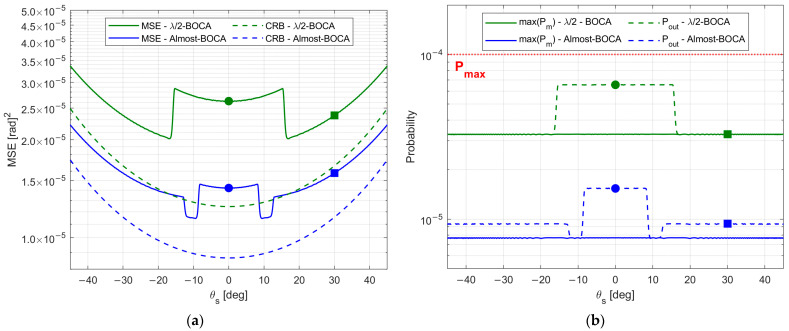
Theoretical performance of the BOCA and the λ/2-BOCA solutions: (**a**) MSE vs. θ; (**b**) maximum pairwise error probability vs. θ.

**Figure 23 sensors-23-04872-f023:**
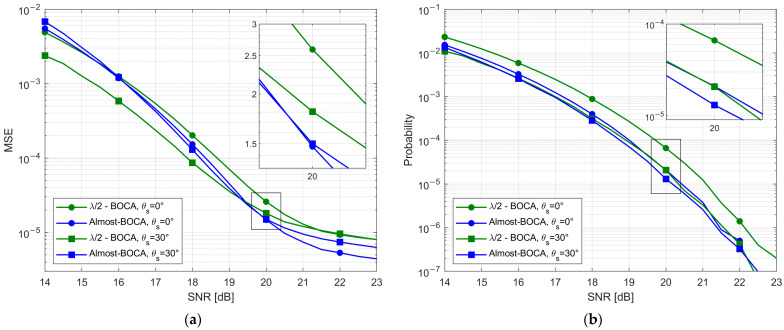
Experimental results (**a**) MSE vs. SNR; (**b**) $P_{out}$ vs. SNR.

**Table 1 sensors-23-04872-t001:** Minimum distance $min\left(S_{pq}\right)$ for the array configurations shown in Figure 4a and the corresponding maximum pairwise error probability $\underset{m}{\max}P_{m}$.

		$z_{1}$	$z_{2}$	$z_{3}$	$z_{4}$
ρ=0.7	$\underset{m}{\max}P_{m}$	10^−20^	10^−20^	10^−20^	2 10^−5^
	$min\left(S_{pq}\right)$	170.25	170.25	170.25	170.25
ρ=0.8	$\underset{m}{\max}P_{m}$	10^−20^	10^−20^	10^−14^	0.2
	$min\left(S_{pq}\right)$	170.25	170.25	170.25	15.48
ρ=0.9	$\underset{m}{\max}P_{m}$	10^−20^	10^−20^	10^−7^	0.2
	$min\left(S_{pq}\right)$	170.25	170.25	170.25	15.48
ρ=1.0	$\underset{m}{\max}P_{m}$	10^−20^	10^−20^	10^−2^	0.2
	$min\left(S_{pq}\right)$	170.25	170.25	170.25	15.48
ρ=1.1	$\underset{m}{\max}P_{m}$	10^−20^	10^−20^	0.2	0.2
	$min\left(S_{pq}\right)$	170.25	170.25	15.48	15.48
ρ=1.2	$\underset{m}{\max}P_{m}$	10^−20^	10^−14^	0.2	0.2
	$min\left(S_{pq}\right)$	170.25	170.25	15.48	15.48
ρ=1.3	$\underset{m}{\max}P_{m}$	10^−20^	10^−9^	0.2	0.2
	$min\left(S_{pq}\right)$	170.25	170.25	15.48	15.48

**Table 2 sensors-23-04872-t002:** Selected case studies, BOCA and λ/2 -BOCA solutions, and achieved ratio $R\left(\theta_{opt},\theta_{int}\right)$.

	$\theta_{max}$	$\theta_{int}$	$\theta_{opt}$	$z_{\lambda /2-BOCA}$	$z_{BOCA}$	$R\left(\theta_{opt},\theta_{int}\right)$
Case study A1	90°	30°	6°	[0.00, 1.00, 4.50]λ	[0.00 1.30 5.84]λ	1.6838
Case study A2			18°	[0.00, 1.00, 4.50]λ	[0.00 0.66 5.22]λ	1.2561
Case study A3			30°	[0.00, 0.50, 4.00]λ	[0.00 0.66 5.26]λ	1.7253
Case study B1	30°	30°	15°	[0.00, 1.00, 4.50]λ	[0.00 0.98 7.74]λ	2.7682
Case study B2			30°	[0.00, 0.50, 4.00]λ	[0.00 0.94 7.82]λ	3.2847

## Data Availability

The data presented in this study are available on request from the corresponding author.

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
