# Peer review of "A Flexible Design Strategy for Three-Element Non-Uniform Linear Arrays"

_sensors, 2023, doi:10.3390/s23104872_

Round 1

Reviewer 1 Report

1. In the Abstract section, some abbreviations are unnecessary.

2. In the received vector model, the steering vector is defined as s(u), which is the source signal?

3. The variable writing in the paper needs to be unified. For example, the writing formats of matrices, vectors, and general variables should be unified respectively.

4. The quality of the language is very poor, and should be improved; The authors should not only revise the structure of the manuscript but also proofread it with the help of native speakers. The rigorous manuscript will show at least respect for the peer review.

5. The conclusions should be extended and future work should be discussed with care.

6. In addition, the references should be unified to fully adhere to the required format of the journal.

Reviewer 2 Report

This paper presents a flexible design strategy for a Non-Uniform Linear Array (NULA) of three sensors for Direction of Arrival (DoA) estimation. NULA provides spatial diversity resulting in satisfactory DoA estimation with limited receiving elements, making it suitable for low-cost passive location applications. The design procedure includes an algorithm to identify the admissible pairwise probability region, which can be modified to incorporate practical design constraints. The proposed design strategy is compared to the ?/2-BOCA solution presented in [14], and it outperforms it. Experimental data verifies the effectiveness of the optimized array.

To improve the quality of their work, authors should avoid limiting their comparisons to a single reference 1976 report. Instead, they are encouraged to broaden their scope and consider the latest literature available for discussion and comparison. This would enhance the relevance and validity of their findings.

Moreover, the authors should explain the experiment and divide it into well-structured sub-sections that clearly present the data and corresponding results. In addition, it is advisable to revise the language and correct any grammatical errors.

Reviewer 3 Report

The manuscript titled “A Flexible Design Strategy for Three-Element Non-Uniform 2 Linear Arrays” describes the potential application for DOA.

However, the authors need to address the following comments:

1.      Hom much distance between three sensors is considered to apply MLE?

2.      How the SNR is considered in a scattering environment?

3.      How much distance between three sensors is considered to apply MLE?

4.      How is the SNR considered in a scattering environment?

5. What is the main contribution of this proposed research work?

6. What are the disadvantages of existing systems? How is the proposed research work superior to existing work?

7. What are the potential applications of the proposed research work?

8. Kindly provide the comparison table showing your potential contribution.

9. Is the proposed work suitable when arrays are placed circularly?

10. While considering DOA, what is the effect of scattering signals?

11. Is CSI at receiver considered?

Round 2

Reviewer 1 Report

The authors have addressed all issues pointed in the previous review.

Reviewer 2 Report

The authors have adequately addressed all of my comments. The paper is well-structured and appropriately formatted. However, minor English language usage and style adjustments are needed to further enhance the paper's quality.